# Integrated pharmaco-proteogenomics defines two subgroups in isocitrate dehydrogenase wild-type glioblastoma with prognostic and therapeutic opportunities

Sejin Oh [1,2,21], Jeonghun Yeom [3,4,5,21], Hee Jin Cho[6,7,21], Ju-Hwa Kim[8], Seon-Jin Yoon [2,9], Hakhyun Kim [2], Jason K. Sa [10], Shinyeong Ju [3,11], Hwanho Lee [2,12], Myung Joon Oh[1], Wonyeop Lee[13], Yumi Kwon[3,11], Honglan Li[13,14], Seunghyuk Choi [13], Jang Hee Han[1,15], Jong Hee Chang [16], Eunsuk Choi [6,17], Jayeon Kim[6,7], Nam-Gu Her[6], Se Hoon Kim [18], Seok-Gu Kang [15,16], Eunok Paek [13✉], Do-Hyun Nam[6,17,19✉], Cheolju Lee [3,4,20✉] & Hyun Seok Kim [1,2✉]

The prognostic and therapeutic relevance of molecular subtypes for the most aggressive isocitrate dehydrogenase 1/2 (*IDH*) wild-type glioblastoma (GBM) is currently limited due to high molecular heterogeneity of the tumors that impedes patient stratification. Here, we describe a distinct binary classification of *IDH* wild-type GBM tumors derived from a quantitative proteomic analysis of 39 *IDH* wild-type GBMs as well as *IDH* mutant and low-grade glioma controls. Specifically, GBM proteomic cluster 1 (GPC1) tumors exhibit Warburg-like features, neural stem-cell markers, immune checkpoint ligands, and a poor prognostic biomarker, FKBP prolyl isomerase 9 (*FKBP9*). Meanwhile, GPC2 tumors show elevated oxidative phosphorylation-related proteins, differentiated oligodendrocyte and astrocyte markers, and a favorable prognostic biomarker, phosphoglycerate dehydrogenase (*PHGDH*). Integrating these proteomic features with the pharmacological profiles of matched patient-derived cells (PDCs) reveals that the mTORC1/2 dual inhibitor AZD2014 is cytotoxic to the poor prognostic PDCs. Our analyses will guide GBM prognosis and precision treatment strategies.

[1] Severance Biomedical Science Institute, Yonsei University College of Medicine, Seoul, Korea. [2] Brain Korea 21 PLUS Project for Medical Science, Yonsei University College of Medicine, Seoul, Korea. [3] Center for Theragnosis, Korea Institute of Science and Technology, Seoul, Korea. [4] Division of Bio-Medical Science & Technology, KIST School, Korea University of Science and Technology, Seoul, Korea. [5] Convergence Medicine Research Center, Asan Institute for Life Sciences, Seoul, Korea. [6] Institute for Refractory Cancer Research, Samsung Medical Center, Seoul, Korea. [7] Precision Medicine Research Institute, Samsung Medical Center, Seoul, Korea. [8] Graduate Program for Nanomedical Science, Yonsei University, Seoul, Korea. [9] Department of Biochemistry and Molecular Biology, Yonsei University College of Medicine, Seoul, Korea. [10] Department of Biomedical Sciences, Korea University College of Medicine, Seoul, Korea. [11] Department of Life Science and Research Institute for Natural Sciences, Hanyang University, Seoul, Korea. [12] Department of Systems Biology, College of Life Science and Biotechnology, Yonsei University, Seoul, Korea. [13] Department of Computer Science, Hanyang University, Seoul, Korea. [14] School of Computer Science and Engineering, Soongsil University, Seoul, Korea. [15] Department of Medical Science, Yonsei University Graduate School, Seoul, Korea. [16] Department of Neurosurgery, Brain Tumor Center, Severance Hospital, Yonsei University College of Medicine, Seoul, Korea. [17] Department of Neurosurgery, Samsung Medical Center, Sungkyunkwan University School of Medicine, Seoul, Korea. [18] Department of Pathology, Yonsei University College of Medicine, Seoul, Korea. [19] Department of Health Sciences and Technology, SAIHST, Sungkyunkwan University, Seoul, Korea. [20] Department of Converging Science and Technology, KHU-KIST, Kyung Hee University, Seoul, Korea. [21]These authors contributed equally: Sejin Oh, Jeonghun Yeom, Hee Jin Cho. ✉email: eunokpaek@hanyang.ac.kr; nsnam@skku.edu; clee270@kist.re.kr; hsfkim@yuhs.ac

Glioblastoma multiforme (GBM) is a lethal form of adult brain cancer with a median overall survival time of 12–15 months[1]. Currently available therapeutic procedures—namely, gross total resection, followed by a combination of radiotherapy and chemotherapy with DNA-alkylating temozolomide[2,3]—are largely ineffective. Indeed, 90% of patients treated according to the standard clinical procedures experience tumor recurrence within 6–9 months after initial treatment[4].

Much research has been conducted using high-dimensional molecular data in an attempt to locate major oncogenic events and therapeutically actionable intervention points and to classify GBM patients into diagnostic and prognostic subgroups[5–7]. The vast majority of these efforts have been confined to the genomic, transcriptomic, and epigenetic level. For example, a notable study performed by The Cancer Genome Atlas (TCGA) group involved sequencing several hundred GBM specimens, from which three major oncogenic signaling pathways were identified: receptor tyrosine kinase/RAS/PI3K, p53 and RB[7]. Based on the expression of 840 genes, the researchers classified GBM into four distinct subtypes: classical (EGFR amplification and CDKN2A deletion), mesenchymal (NF1 deletion and expression of mesenchymal markers), proneural (PDGFRA amplification, IDH1 mutation and expression of proneural development genes), and neural (expression of neuronal markers)[8]. Despite these efforts, these mutation-based and transcriptome-based approaches have found limited clinical application, and only a few biomarkers, including IDH mutation (favorable prognoses, secondary GBM)[9], MGMT promoter methylation (benefit from temozolomide)[10], and 1p/19q co-deletion (chemosensitivity)[11] are being used in clinic. Meanwhile, IDH wild-type GBM, which is found in ~90% of all GBM cases, represents the most aggressive glioma subtype[12]. Establishing predictive biomarkers or patient stratification strategies for use in developing targeted therapies and identifying determinants of long-term survival of IDH wild-type GBM remain challenges.

In this regard, proteogenomic studies in other cancers have demonstrated that DNA-level and RNA-level alterations are insufficient to predict protein activity[13–15]. Therefore, proteome-based patient stratification might provide a more effective approach with which to predict prognosis and susceptibility to targeted agents. However, although several studies have conducted proteomic analysis of glioma tissue samples[16,17] or secreted proteins in blood[18], large-scale proteomic characterization in the context of GBM has not yet been reported.

Here, we delineate GBM tumors based on proteome data and identify prognostic and therapeutic biomarkers particularly for IDH wild-type GBM. We generate global-proteomic and phospho-proteomic data for a panel of 50 glioma tissues (39 IDH wild-type GBMs) with previously annotated genomic, transcriptomic, and clinical information as well as the responses of matched neurosphere-like patient-derived cells (PDCs) to targeted therapies. Our integrated pharmaco-proteogenomic approach provides insight into GBM intertumoral and intratumoral heterogeneity in cell of origin, oncogenic signaling, and metabolic pathways. Our data highlight potentially effective prognostic and therapeutic strategies for IDH wild-type GBM patients.

## Results

**Proteomic data represent glioma disease state.** To gain insight into GBM at the proteomic level, we assembled 39 IDH wild-type GBM samples, along with two IDH mutant GBM and nine low-grade glioma (LGG) samples as a control, from the Samsung Medical Center (SMC) cohort, for which pre-existing whole-exome sequencing (WES) and RNA sequencing (RNA-seq) data already exist[19]. These samples displayed broad coverage of major driver mutations[5], including EGFR, EGFRvIII (deletion in exon 2–7), TP53, RB1, PTEN, and PIK3CA (Fig. 1a), and copy number alterations (CNAs) in CDKN2A/TP53 (deletion) and EGFR/PDGFRA (amplification) (Supplementary Fig. 1a), indicating that these samples represented the GBM mutational spectrum. The samples also spanned all four RNA subtypes[8] (Supplementary Fig. 1b). 20 out of 50 samples were obtained redundantly from multiple regions or at different time points and had different properties regarding mutation, RNA subtype, 5-aminolevulinic acid (5-ALA) positivity, location (locally adjacent or core and margin of tumors), or primary/relapse status (Supplementary Data 1). Unsupervised clustering showed that samples from the same patient showed a high degree of DNA-level similarity (Fig. 1a).

We first measured the global proteome and phosphoproteome levels relative to pooled global internal standards (GIS) by isobaric labeling with six-plex tandem mass tag (TMT) reagents followed by liquid tandem mass spectrometry (LC–MS/MS). We obtained measurements for all 50 tumor samples and four normal brain tissues by randomly assigning them to 11 TMT sets (Fig. 1b). After removing isoforms, we quantified 9367 protein groups and the phosphorylation of 8020 amino acid sites: each TMT set possessed an average of 6294 protein groups and 2796 phosphorylation sites (Supplementary Fig. 1c). We then selected for further analysis 3909 protein groups that were quantified in all GIS and localized to all cellular compartments (Supplementary Fig. 1d) and 4489 phospho-sites quantified in three or more GIS (Supplementary Data 2). Significant overlap between the single amino acid variants (SAVs) detected in this study by LC–MS/MS and the previously annotated single nucleotide variants (SNVs; Fig. 1c) indicated that our proteomic assay could successfully detect mutant proteins. In addition, the protein expression levels were generally positively correlated with the RNA levels, except for genes involved in certain housekeeping functions, including genes associated with ribosomes and oxidative phosphorylation (OXPHOS—the mitochondrial process through which ATP is synthesized via the electron transport chain coupled to substrate oxidation; Fig. 1d), which is concordant with the findings of previous studies on other tumor types[13–15].

We next established the degree of similarity between samples at the proteomic level by unsupervised hierarchical clustering. Intriguingly, matched samples were clustered primarily by RNA subtypes ($P < 0.001$; permutation test) or clinical phenotypes, such as 5-ALA positivity, tumor grade and primary/relapse status ($P < 0.05$; permutation tests) (Fig. 1e), instead of being clustered by DNA-level similarity. This result suggests that proteome might better represent the disease state and underlying biology than genome.

Given the high quality of the proteome data, we were able to identify differentially expressed proteins by comparing glioma samples with adjacent normal tissue samples. As expected, compared to normal tissues, gliomas showed elevated levels of proteins involved in cell proliferation and immune responses (Supplementary Fig. 1e; Supplementary Data 3). We then compared IDH mutant ($N = 6$; 2 grade IV and 4 low grade) and IDH wild-type ($N = 44$; 39 grade IV and 5 low grade) gliomas, because the latter are associated with a significantly worse prognosis than the former. We found that IDH wild-type gliomas were significantly associated with elevated phosphorylation levels at 25 phospho-sites (Supplementary Data 3). Of these sites, the phosphorylation of signal transducer and activator of transcription 1 (STAT1) at serine-727, which is a marker of STAT1 activation[20], correlated with elevated target protein levels in IDH wild-type tumors (Fig. 1f). Also, IDH mutation status directly affected pSTAT1-S727 levels, as shown in an IDH

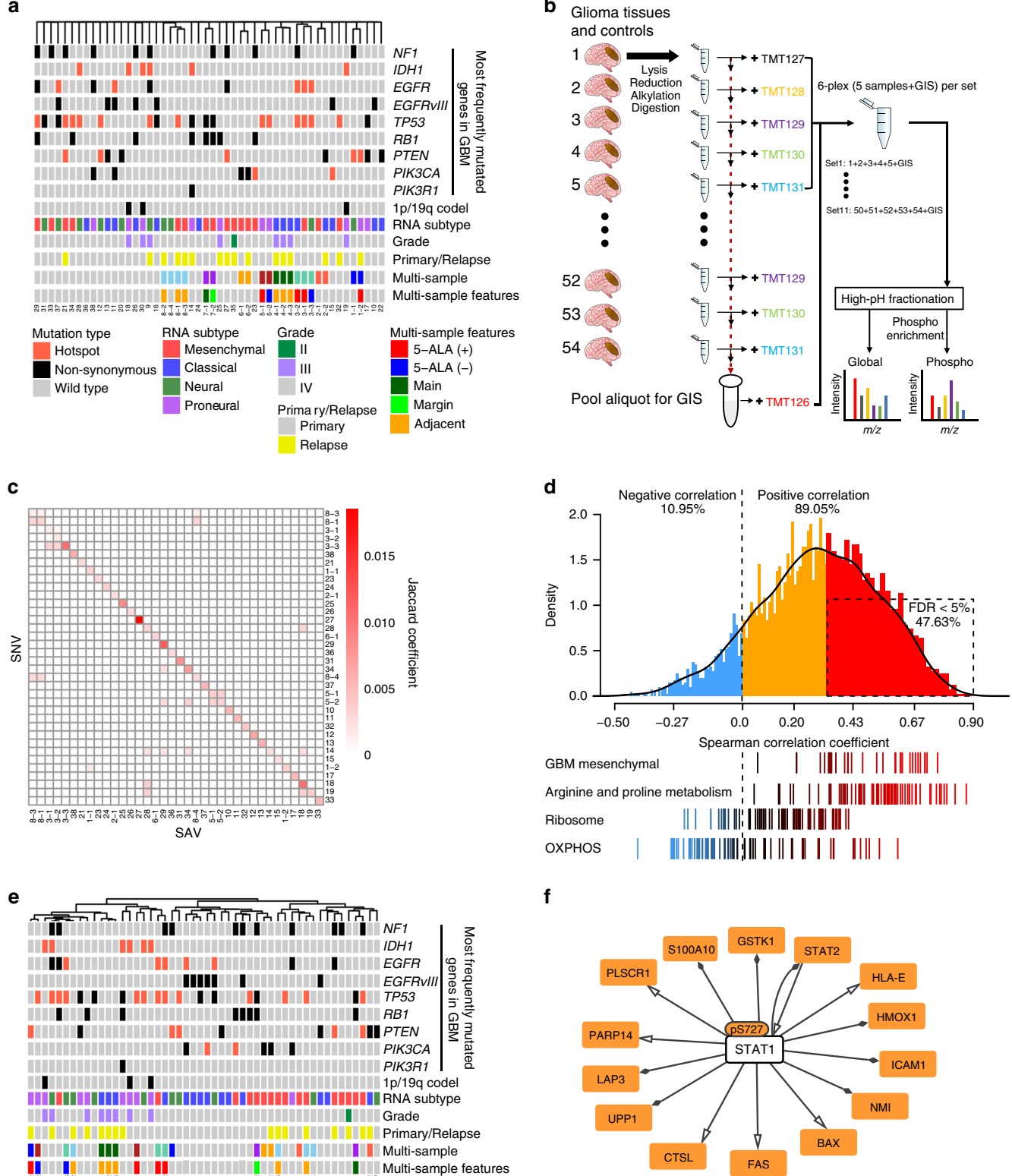

wild-type and mutant GBM cell line pair in an isogenic U87MG background (Supplementary Fig. 1f). Collectively, our results suggest that quantitative proteomic data can accurately cluster glioma samples by clinical phenotype and identify activated pathways and their regulators.

**Two proteomic subgroups of *IDH* wild-type GBM.** We next classified the tumors based on the proteome data. A consensus clustering algorithm analysis of the 3909 protein groups detected in all the tumor samples identified two stable proteomic subtypes for *IDH* wild-type GBM: glioblastoma proteome cluster 1 (GPC1, $N = 26$) and GPC2 ($N = 13$) (Fig. 2a). Inclusion of *IDH* mutant GBMs and LGGs did not alter the classification of *IDH* WT GBM tumors (Supplementary Fig. 2a): the two *IDH* mutant GBMs were classified with GPC2 tumors, and the nine LGGs were significantly associated with GPC2 tumors ($\chi^2$ test $P = 0.0029$).

**Fig. 1 Proteomic characterization reveals inter- and intra-patient molecular heterogeneity of glioblastoma multiforme (GBM).** **a** Characteristics of *IDH* wild-type GBM (*N* = 39), *IDH* mutant GBM (*N* = 2) and low-grade glioma (*N* = 9) tissue samples. Unsupervised hierarchical clustering with complete linkage was used to cluster samples based on the 1 – Jaccard coefficient as the distance metric. The type of mutations in the 8 most frequently mutated GBM genes[5] are color-coded according to the legend. The multi-sample row displays multiple tumor samples obtained from the same patient as the same color; no color indicates unique samples. 5-ALA (within multi-sample features) indicates the intensity of the 5-aminolevulinic acid-induced fluorescence. **b** Overview of the multiplexed quantitative proteomic assay of glioma tissues. Trypsin-digested glioma (*N* = 50) and control normal tissues (*N* = 4) were tagged with a six-plex tandem mass tag (TMT): TMT127-131 for samples and TMT126 for the global internal standard (GIS) control. A total of 11 sets for 54 samples were prepared. High-pH fractionated peptides were subjected to liquid chromatography-tandem mass spectrometry to identify and quantify phosphopeptides and global proteins. See "Methods" for further details. **c** Coherence map of single-nucleotide variants (SNV) and single amino acid variants (SAVs). **d** Correlations between mRNA and protein levels in glioma tissue samples. (Top) Density plot of Spearman's correlation coefficients between mRNA and protein abundance using the 8034 proteins detected in all GIS samples (*N* = 4071 at the gene level). Statistically significant positive correlations with a false discovery rate (FDR) <5% are indicated by the dashed-line box. (Bottom) Distribution of correlation coefficients for gene sets of interest. **e** Unsupervised hierarchical clustering of the 50 samples with global-proteomic data. Complete linkage and the distance metric 1 – Pearson's correlation coefficient was used for clustering. **f** Genetic regulatory network activated in *IDH* wild-type tumors. The transcription factor–target gene regulatory network was formed by the significantly upregulated phosphoproteins and global proteins in *IDH* wild-type tumors using the OmniPath database[61] in Cytoscape. Source data are provided as a Source Data file.

Removing non-unique samples (i.e., any sample that shared a patient of origin) or taking all the GBMs regardless of *IDH* genotype did not affect the binary nature of the classification (Supplementary Fig. 2b–d). The stability of this classification supports the robustness of the subtypes and implies that binary classification is applicable to GBM in general.

The GPC1 global proteome expression pattern was distinct from that of normal brain tissues, whereas the GPC2 global proteome expression displayed a normal brain tissue-like pattern (Fig. 2b, top panel). However, the purities of GPC2 tumor did not significantly differ from those of GPC1 based on comparable variant allele frequency (VAF) distribution (Fig. 2b, bottom panel), which suggests that the abundance of normal cells in tumors does not contribute to these proteomic differences. A subtype switch, likely driven by therapeutic treatment, was observed in recurrent tumors, with two of the three samples switching from GPC1 into GPC2 (Supplementary Fig. 2e). This finding demonstrates protein-level subtype plasticity, which is in agreement with previous findings from a longitudinal transcriptome analysis[21].

The two GPC subtypes were largely independent of the four RNA subtypes[8] ($\chi^2$ test *P* = 0.122; Fig. 2c). Notably, *EGFRvIII* and *PIK3CA* mutations were exclusively found in GPC1 tumors, whereas other GBM driver mutations in *TP53*, *NF1*, *PTEN*, *RB1*, and *EGFR* (non-*vIII*) were relatively evenly distributed between the two subtypes (Fig. 2d). These results suggest that *IDH* wild-type GBM can be classified into two stable protein subtypes that are distinct from RNA subtypes.

**OXPHOS-related proteins determine proteomic subtypes**. To identify the key proteins that characterize the GPC subtypes, we conducted a principal component analysis (PCA) of the global proteome data. The first principal component (PC1) successfully stratified the two GPCs (Fig. 3a; left panel). A gene ontology analysis of the top 10% of proteins with the highest absolute PC1 loading values revealed a notable enrichment of proteins involved in OXPHOS (Fig. 3a; right panel and Supplementary Fig. 3a). Moreover, the expression of OXPHOS-related proteins was significantly lower in GPC1 than in GPC2. Importantly, we found no differences in the OXPHOS mRNA levels between the subtypes (Fig. 3b).

Compared to normal cells, GPC2 tumors expressed similar levels of proteins involved in glycolysis, serine biosynthesis, the tricarboxylic acid (TCA) cycle, glutaminolysis, and OXPHOS, but at significantly higher levels than those found in GPC1 tumors (Fig. 3b, c). This similarity indicates that GPC2 tumors primarily generate ATP via OXPHOS, in a similar manner to normal cells under aerobic conditions.

Compared with GPC2 tumors, GPC1 tumors expressed higher levels of lactate dehydrogenase A (LDHA) and proteins involved in glucose uptake, hexokinase 2 (HK2), the pentose phosphate pathway (PPP), and the one-carbon pathway (Fig. 3c). Notably, GPC1 expressed higher levels of pyruvate kinase m2 (PKM2) and HK2, whereas GPC2 had an elevated PKM1 (Fig. 3d; Supplementary Fig. 3b). In addition, GPC1 tumors exhibited elevated expression of IDH1 protein (the primary producer of NADPH in GBM)[22,23] beyond the levels found in GPC2 tumors (Supplementary Fig. 3c). The GPC1-activated PPP and one-carbon pathway also generate NADPH, which is a reducing equivalent for tumor cells affected by the Warburg effect[24]. Coherent with our proteomic data, GBM cell lines belonging to gene expression-based surrogate-GPC1 subtype (sGPC1) exhibited higher lactate levels (Supplementary Fig. 3d) in the analysis of cancer cell line encyclopedia metabolomics data[25]. Together, these results suggest that GPC1 tumors metabolically rely on the Warburg effect.

**GPC-subtype-dependent expression of cell-of-origin markers**. Neural stem cell (NSC) is considered a cell of origin of GBM[26]. To understand whether each GPC subtype has a distinct cellular origin, we compared the levels of NSCs (Nestin, Vimentin, CD44), oligodendrocytes (OSP, MOG), and astrocytes (GLUL, GLT-1, GLAST, HepaCAM, ALDH1A1, S100β) marker proteins that were detected in all GIS. GPC1 tumors had significantly elevated Nestin, Vimentin, and CD44 (Fig. 4a). In GPC1 tumors, we also observed elevations in an active form of cortactin phosphorylated at T364/S368/T401/S405 and its interacting partner, Arp2/3 complex subunits (Fig. 4b; Supplementary Data 3), which are components not only of filopodia/lamellipodia (cytoplasmic protrusions of migratory cells) but also of invadopodia (invasive protrusions of transformed cells). By contrast, GPC2 tumors significantly overexpressed the oligodendrocyte and astrocyte markers OSP, MOG, GLT-1, GLAST, and HepaCAM (Fig. 4a). GLUL, ALDH1A1, and S100β were also relatively highly expressed in GPC2 tumors, but were marginally insignificant (Fig. 4a). Cumulative evidence indicates that GBM stem cells are immune-resistant[27]. Concordantly, GPC1 had elevated levels of *CD274* (PD-L1, two-way ANOVA *P* = 2.02E−6) and *PDCD1LG2* (PD-L2, two-way ANOVA *P* = 4.75E−13) (Supplementary Fig. 4a). Together, with the observation that recurrent tumors tend to be GPC2 (Supplementary Fig. 2e), one potential explanation of these data is that GPC1 tumors originate from NSCs, whereas GPC2 tumors differentiate from GPC1. However, we

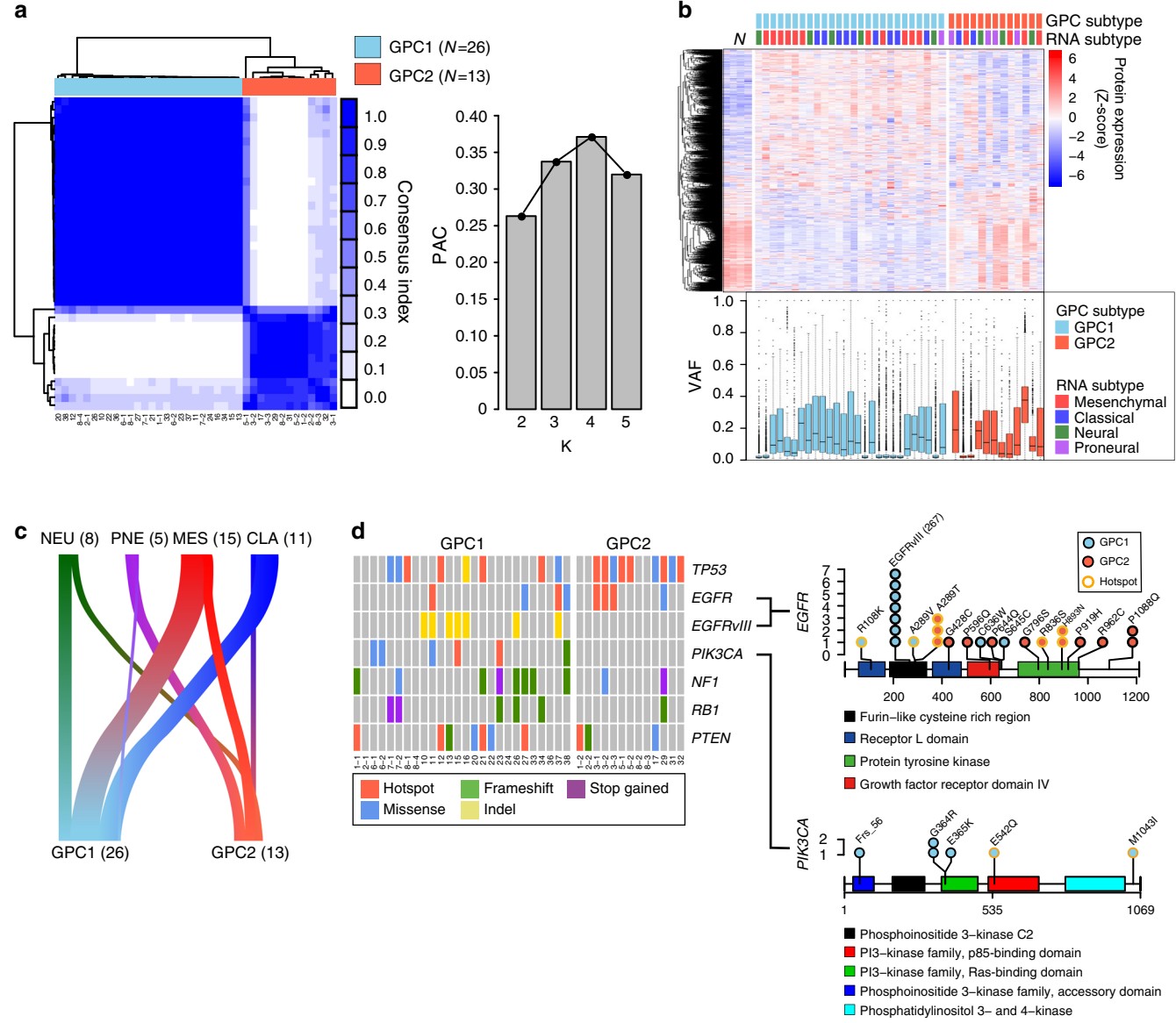

**Fig. 2 Proteomic subtypes of *IDH* wild-type GBM associate with clinical and genomic features. a** Consensus clustering of *IDH* wild-type GBM samples (*N* = 39) based on global proteome data using 1 – Pearson's correlation coefficient as the distance metric. (Left) Heatmap of the consensus score. (Right) The bar chart indicates the proportion of ambiguous clustering (PAC) for the indicated K values. The number of clusters (K) with the lowest PAC score is considered the optimal cluster number. **b** Characteristics of proteomic subtypes. The columns represent the samples grouped by proteomic subtype. The clustering heatmap represents the *Z*-score-normalized protein expression levels. The box-and-whisker plot represents the medians (middle line), first quartiles (lower bound line), third quartiles (upper bound line), and the ±1.5× interquartile ranges (whisker lines) of the variant allele frequency (VAF) of single nucleotide variants (SNVs)[19] per sample. N: normal control samples. **c** The river plot illustrates the association between proteomic and RNA subtypes. The width of the edge between two subtypes is shown in proportion to the number of corresponding samples. CLA classical, PNE proneural, NEU neural, MES mesenchymal. Correlation between the two subtyping methods was insignificant by the chi-square test (*P* = 0.122). **d** GBM driver mutations associated with proteome subtypes. (Left) The color-coded matrices indicate the cBioPortal-annotated mutation types in the samples grouped by proteomic subtype for the six frequently mutated genes in GBM. (Right) Somatic mutations in *EGFRvIII* and *PIK3CA* that were exclusively found in GPC1 and GPC2 tissues. The *x* axis indicates the position in the gene, and the *y* axis indicates the frequencies of the mutations. The color in the circle indicates a mutation found either in a GPC1 (blue) or in a GPC2 (red) sample. The outline color of the circle indicates the hotspot mutations. Rectangles with different colors represents protein domains. Source data are provided as a Source Data file.

cannot exclude the possibility that GPC2 tumors may originate directly from oligodendrocytes and astrocytes.

**PHGDH predicts favorable prognosis in *IDH* wild-type GBM.** A previous study identified GBM prognostic gene expression biomarkers comprising 66 favorable and 205 unfavorable genes[28], however, protein level validation has yet been pursued. Univariate Cox regression analysis of our GBM cohort revealed that only 11

of these markers were reproduced at the protein level (Supplementary Data 4), and of those, we identified two biomarkers as favorable for *IDH* wild-type GBM (phosphoglycerate dehydrogenase, PHGDH, and Raftlin family member 2, RFTN2), and one as unfavorable (FKBP prolyl isomerase 9, FKBP9; Fig. 4c, d). Importantly, the protein expression of all three of these markers differed between the two GPC subtypes—FKBP9 was elevated in GPC1, whereas PHGDH and RFTN2 were higher in GPC2

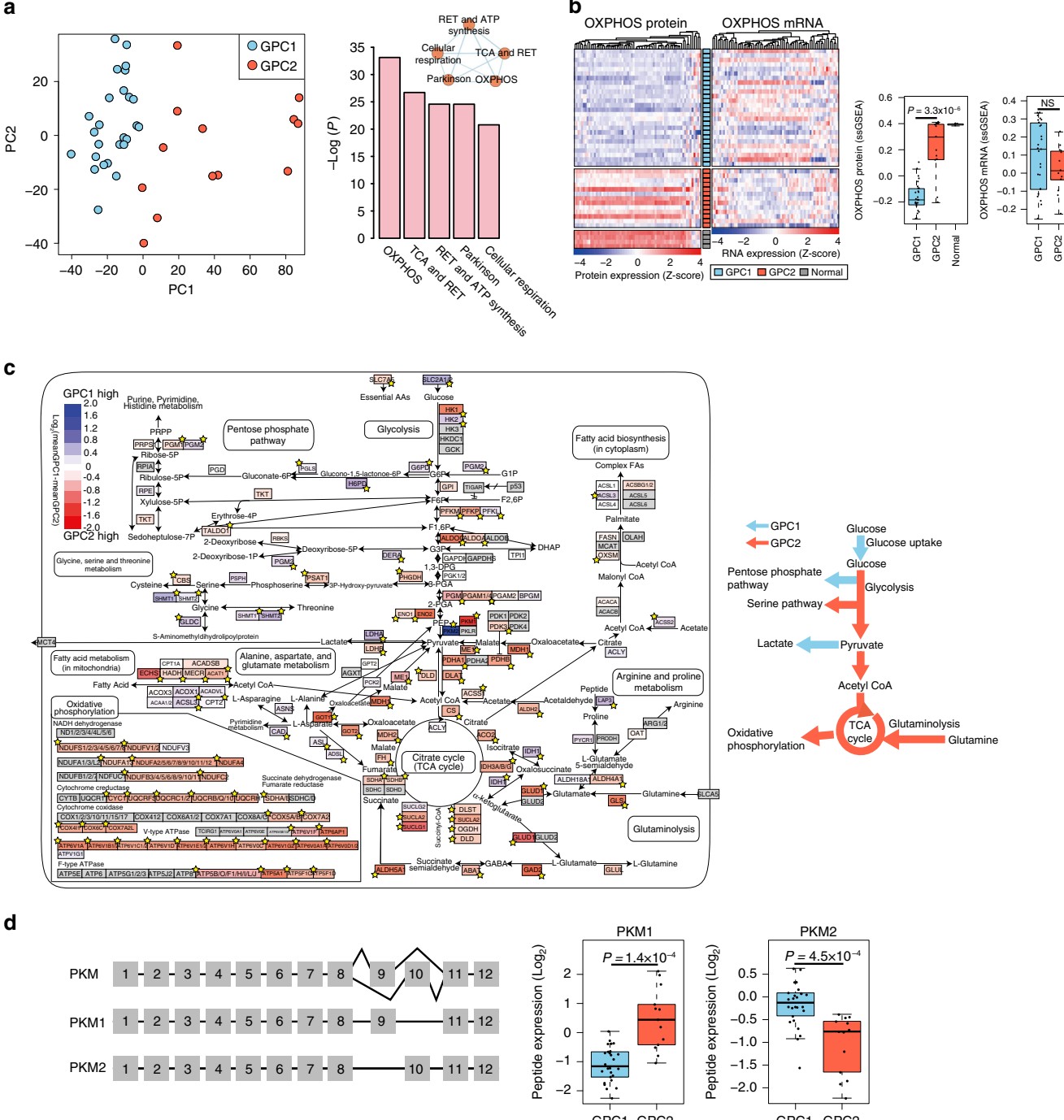

**Fig. 3 Proteins involved in central carbon metabolism are major determinants of *IDH* wild-type GBM proteomic subtypes. a** Principal component analysis (PCA) of 39 *IDH* wild-type GBM samples using global proteome data (left). The statistical significance of the enriched gene sets with the top 10% loading genes (N = 390) in PC1 is shown in the right panel. The mutual similarity of the top five gene sets is presented using EnrichmentMap with default parameters. RET respiratory electron transport. **b** Clustering heatmaps of *IDH* wild-type GBM tissues with the OXPHOS-related genes based on protein and mRNA expression data (left). The rows represent the tumor samples grouped by proteomic subtype, and the columns represent the genes belonging to the OXPHOS-related gene set. Comparisons of the single-sample gene set enrichment analysis (ssGSEA) scores for both proteins and genes are shown on the right. The box-and-whisker plots represent the medians (middle line), first quartiles (lower bound line), third quartiles (upper bound line), and the ±1.5× interquartile ranges (whisker lines); the raw data are overlaid. *P*-values were calculated using the two-sided unpaired Wilcoxon rank-sum test. NS not significant. The number of samples for GPC1, GPC2, and normal is 26, 13, and 4, respectively. **c** Pathway view of differentially expressed proteins involved in central carbon metabolism between the two proteome subtypes. The stars indicate a statistically significant difference in protein expression between the two GPC subtypes (*P* < 0.05; two-sided unpaired Student's *t*-test. See Supplementary Data 3 for exact *P*-values). Genes with no available protein expression data are shown in gray. The elevated pathways in GPC1 and GPC2 tumors are summarized as colored arrows in the right panel. **d** Alternative splicing of pyruvate kinase muscle isozyme (PKM) isoforms (left) and comparison of the peptide expression of PKM1 and PKM2 in GPC subtypes (right). The *y* axes of the boxplots represent the peptide expression of PKM isoforms normalized by the GIS. The description of box-and-whisker plots are the same as in **b**. The *P*-value was calculated by two-sided unpaired Student's *t*-test. The number of samples for GPC1 and GPC2 is 26 and 13, respectively.

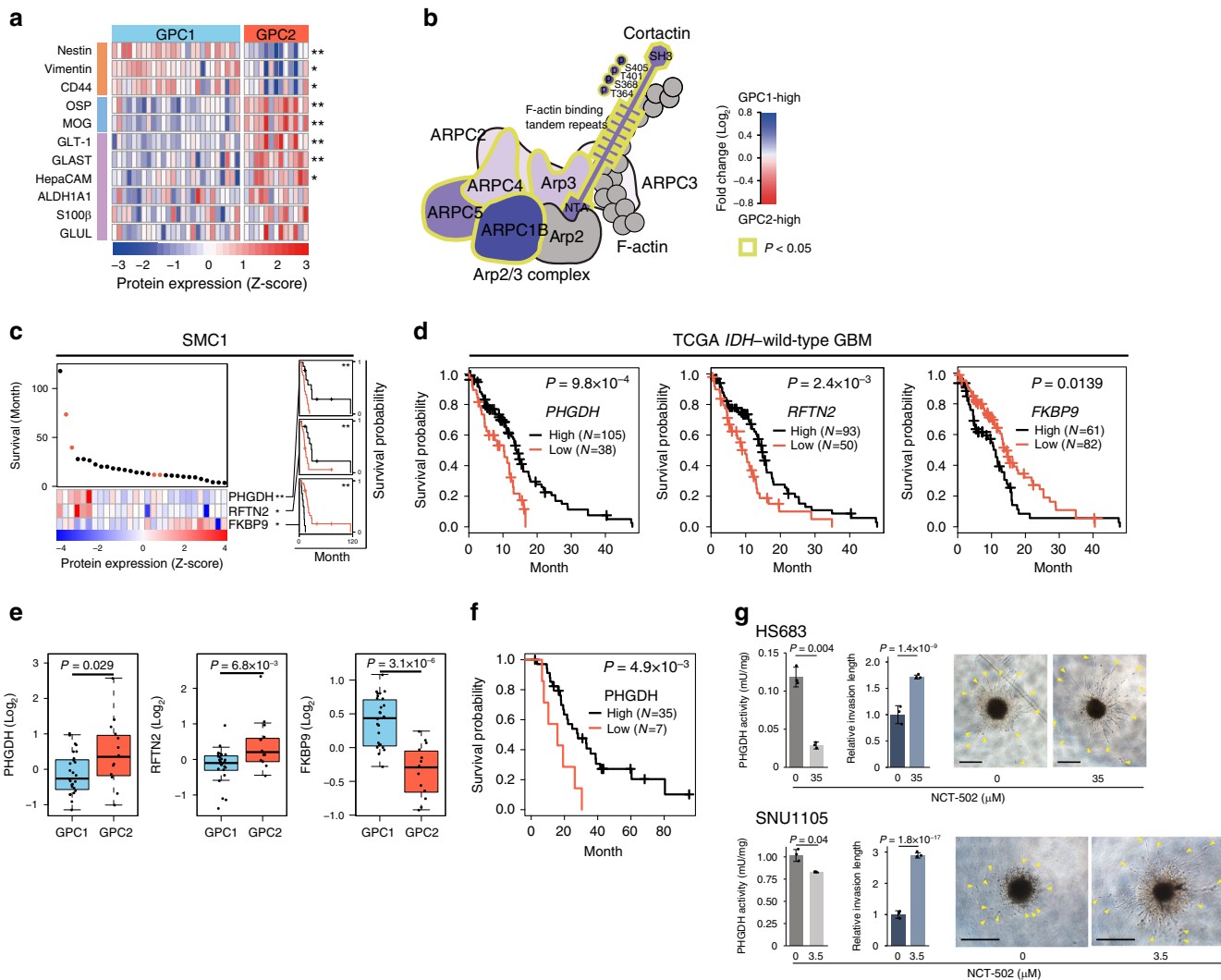

**Fig. 4 GPC2-associated PHGDH predicts a favorable prognosis in *IDH* wild-type GBM. a** Differential expression of protein markers for NSCs (orange), oligodendrocytes (blue), and astrocytes (purple) between proteomic subtypes. *P < 0.05, **P < 0.001; two-sided unpaired Student's *t*-test. See Supplementary Data 3 for exact *P*-values. **b** The cortactin-Arp2/3 complex is elevated in GPC1 tumors. The gray color indicates proteins with no available protein expression data. *P*-values were calculated by two-sided unpaired Student's *t*-test. **c** Prognostic biomarker proteins in *IDH* wild-type GBM. Deceased and surviving patients are denoted by black and red dots, respectively. *P < 0.05, **P < 0.01; univariate Cox regression test. Kaplan–Meier (KM) survival curves for *IDH* wild-type GBM patients (N = 29) in the SMC1 cohort were shown on the right for each of the three proteins (black: high expression, red: low expression). **P < 0.01; two-sided log-rank test. **d** Kaplan–Meier (KM) survival curves. Patients were classified by the optimal gene expression thresholds reported by Uhlen et al.[28]. *P*-values were calculated using the two-sided log-rank test. **e** Box-jitter plots for the abundance of the indicated proteins. The description of the box-and-whisker plots is the same as in Fig. 3b. Statistical significance of the downregulation of favorable markers (PHGDH and RFTN2) and upregulation of an unfavorable marker (FKBP9) was evaluated by Student's *t*-test (one-sided unpaired). The number of samples for GPC1 and GPC2 is 26 and 13, respectively. **f** KM survival curves for PHGDH-high vs. low patients in the SMC-TMA cohort. Patients were classified as described in **d**. *P*-values were calculated as in **d**. **g** PHGDH activity (left), relative invasion lengths (middle), and representative images of 3D invasion (right) of the indicated tumor spheres after treatment of vehicle (DMSO) or NCT-502 for 48 h at the indicated concentrations. Two-sided unpaired Student's *t*-test was used to compare PHGDH activity. Two-way ANOVA was used for the comparison of relative invasion length between treatment groups. Error bars indicate ±SD, N = 3. Arrowheads indicate invasive fronts. Scale bar: 500 μm. Source data are provided as a Source Data file.

(Fig. 4e). However, GPC subtypes did not directly show a significant difference in prognosis (two-sided Log-rank test P = 0.0548).

Of the three proteins, PHGDH was the strongest biomarker (univariate Cox P = 0.0071) associated with long-term survivors of *IDH* wild-type GBM patients in the SMC1 cohort (Fig. 4c), as well as in other independent data sets at the mRNA-level (Supplementary Fig. 4b). Favorable prognosis of PHGDH-high tumors was further validated in 42 independent *IDH* wild-type GBM tumors assessed by immunohistochemistry on a tumor tissue microarray (SMC-TMA) using an anti-PHGDH antibody

(Fig. 4f). The good prognosis of the PHGDH-high group suggests a functional role for PHGDH in limiting tumor aggressiveness. Intriguingly, NCT-502, a chemical inhibitor of PHGDH, significantly increased invasion of tumor spheres (Fig. 4g) derived from PHGDH-active GBM cell lines (Supplementary Fig. 4c, d and Supplementary Data 5) into 3D matrix. Conversely, PHGDH overexpression in PHGDH-deficient GBM cell lines decreased invasion (Supplementary Fig. 4e), suggesting that PHGDH may prolong patient survival by suppressing tumor invasion via its increased enzymatic activity. PHGDH, which catalyzes the first step of serine biosynthesis by converting 3-phospho-D-glycerate

to 3-phosphonooxypyruvate, is also known to mediate a promiscuous function with the ability to convert α-ketoglutarate into D-2-hydroxyglutarate (D-2-HG), similar to the *IDH1* mutant protein[29]. In support of this, we found a positive correlation between *PHGDH* levels and 2-HG production (Supplementary Fig. 4f) by analyzing 878 *IDH* wild-type cancer cell-line metabolome data[25].

Taken together, these data indicate that GPC2-associated PHGDH predicts a favorable prognosis in *IDH* wild-type GBM.

**Spatial and cell-type characteristics of GPC subtypes.** Recent single-cell analyses have revealed marked intratumoral heterogeneity in GBM at the transcriptional level. This finding indicates that individual tumors contain single cells with a spectrum of different subtypes and hybrid cellular states, and thus implies that a dominant cellular population might determine the representative subtype of bulk tumors[30]. We therefore aimed to apply our GBM proteomic subtype classification to single-cell transcriptome data reported by Darmanis et al.[31], which comprised 3589 single cells from four patient *IDH* wild-type GBM tumors and provided information regarding the brain cell type (vascular, immune, neuronal, and glial) and three-dimensional location (tumor core, periphery) (see "Methods" and Fig. 5a). Of the 3589 single cells, 357 and 428 cells were classified as sGPC1 and sGPC2, respectively (permutation test $P < 0.05$; Fig. 5a, Supplementary Fig. 5a). The mapping of these cells on t-distributed stochastic neighbor embedding (t-SNE) coordinates revealed a distinct clustering pattern of two proteome subtypes (Fig. 5b). Consistent with our previous findings, NSC markers *CD44* and *VIM* were significantly elevated in single cells of the sGPC1 subtype, whereas oligodendrocyte marker *OLIG2*, *OSP*, and *MOG* and the astrocyte markers *SLC1A2* and *S100B* were elevated in the sGPC2 subtype (Fig. 5c).

According to the annotation by Darmanis et al.[31], the majority of single cells located in the tumor periphery were non-neoplastic cells (95%), whereas the tumor core was largely composed of neoplastic (44%) and myeloid cells (50%). As expected, we found that each of the tumor cores from the four patients comprised a mixture of cells of the two proteome subtypes; however, the ratios of the two subtypes in neoplastic cell populations of the four tumors were highly variable, and made a notable contribution to determining the dominant proteome subtype of the tumor (Fig. 5d). By contrast, the surrounding environment primarily determined the proteomic subtype of normal cell populations: 84% of normal cells in the tumor core displayed sGPC1 features, whereas 70% of normal cells in the periphery displayed sGPC2 features (Fig. 5e). Because the two subtypes showed differential immune evasion characteristics, we further investigated whether the neoplastic single cells of each subtype variably expressed immune checkpoint ligands. Indeed, a PD-1 ligand *PD-L1* was upregulated in the neoplastic cells of sGPC1 tumors (Fig. 5f). To further validate the intratumoral heterogeneity of GPC subtypes at a single cell level, we used a tumor microarray (SMC-TMA) of independent *IDH* wild-type GBM tissues to measure relative expression of PHGDH (good prognostic marker representing GPC2, Fig. 4f) and Nestin (representing GPC1, Fig. 4a) by multiplex fluorescent immunohistochemistry that generated reliable signal intensities at a single cell resolution. Consistent with our findings in the proteomic analysis, sGPC1 tumors exhibited a significantly higher fraction of Nestin-positive neoplastic cells, whereas sGPC2 tumors comprised a significantly higher fraction of PHGDH-positive neoplastic cells (Supplementary Fig. 5b). Intratumoral heterogeneity, observed from the scRNA-seq data, was clearly seen in the multiplex fluorescent immunohistochemistry results. Both Nestin+/PHGDH− cells (representing GPC1 subtype) and PHGDH+/Nestin− cells

(representing GPC2 subtype) were found in all tumor cores, albeit with different ratios matching their sGPC subtypes (i.e., two sGPC1 tumors contained a higher frequency of Nestin+ cells), whereas two sGPC2 tumors contained higher frequency of PHGDH+ cells (Fig. 5g).

These results indicate that GBM tumors comprise cells belonging to both GPC subtypes, and that the ratio of neoplastic cell subtypes influences the overall tumor characteristics. Besides, unlike normal cells in the tumor core which displayed a static sGPC1-like proteomic feature, neoplastic cells in the tumor core exhibited highly variable GPC subtypes, suggesting that the proteomic subtypes of neoplastic cells are largely determined by cancer intrinsic factors rather than the tumor microenvironment.

**GPC-subtype-dependent sensitivities to targeted therapies.** In a previous study, we screened 50 PDCs derived from the same gliomas used here, against 60 anticancer-targeted agents that covered major oncogenic pathways[19]. Of the 60 drugs tested, 51 were cytotoxic to at least one of the PDCs. Here, we determined whether proteome-based patient stratification predicts susceptibility to targeted agents using the PDC data set. Although the PDCs had been cultured in vitro for several passages for the drug assay, a substantial enrichment of statistically significant correlations was observed between protein biomarkers (measured in tumor tissues) and drug–response phenotypes (assayed in PDCs) (Fig. 6a). Known drug targets were most strongly correlated with the drug–response, particularly at the protein level (Supplementary Fig. 6a). For example, bortezomib and panobinostat cytotoxicities were significantly correlated with the protein expression levels of the 20S proteasomal subunits and histone deacetylase (HDAC)1/2, respectively (Fig. 6b), which agrees with previous studies showing that target protein expression levels in cancer cells determine the anticancer activities of proteasome and HDAC inhibitors[32,33]. Interestingly, these correlations were less evident at the mRNA level (Fig. 6b).

We subsequently examined whether any of the 51 targeted agents showed selective cytotoxicity against GPC1 or GPC2 subtype PDCs. Using the median effective dose (ED50) or area under the curve (AUC) values, we identified four GPC1-selective drugs (tandutinib, crizotinib, olaparib, and AZD2014) and two GPC2-selective drugs (erismodegib and canertinib) (Fig. 6c and Supplementary Data 6). Coherent drug-sensitivity and target-pathway activation relationships for all of these drugs were observed at the protein-levels (Fig. 6d): tandutinib (PDGFR inhibitor), PDGFR_Binding; crizotinib (ALK, MET, ROS1 inhibitor), Oncogenesis_by_MET; olaparib (PARP inhibitor), BRCAness score; AZD2014 (mTORC1/2 dual inhibitor), Translational_Initiation; erismodegib (Hedgehog inhibitor), Hedgehog_GLI_Pathway; and canertinib (pan-ERBB inhibitor), ERBB_Pathway. Taken together, these data suggest that tandutinib, olaparib, crizotinib, and AZD2014 might be a promising targeted therapy for GPC1 tumors and that erismodegib and canertinib might be more promising for GPC2 tumors.

**Protein markers inform sensitivities to targeted therapies.** The majority of the 60 targeted agents used in this study inhibit receptor tyrosine kinases (RTKs), the phosphorylation of which serves as an important marker of their activation statuses. Therefore, we endeavored to find phosphoproteomic markers that are associated with specific drug responses using phosphoproteomic data for the 50 gliomas and drug–response data for the matched PDCs (Supplementary Data 6). The left-skewed *p*-value distribution that we obtained from the correlation test indicates the enrichment of statistically significant drug-response and phosphoprotein relationships (Supplementary Fig. 6b).

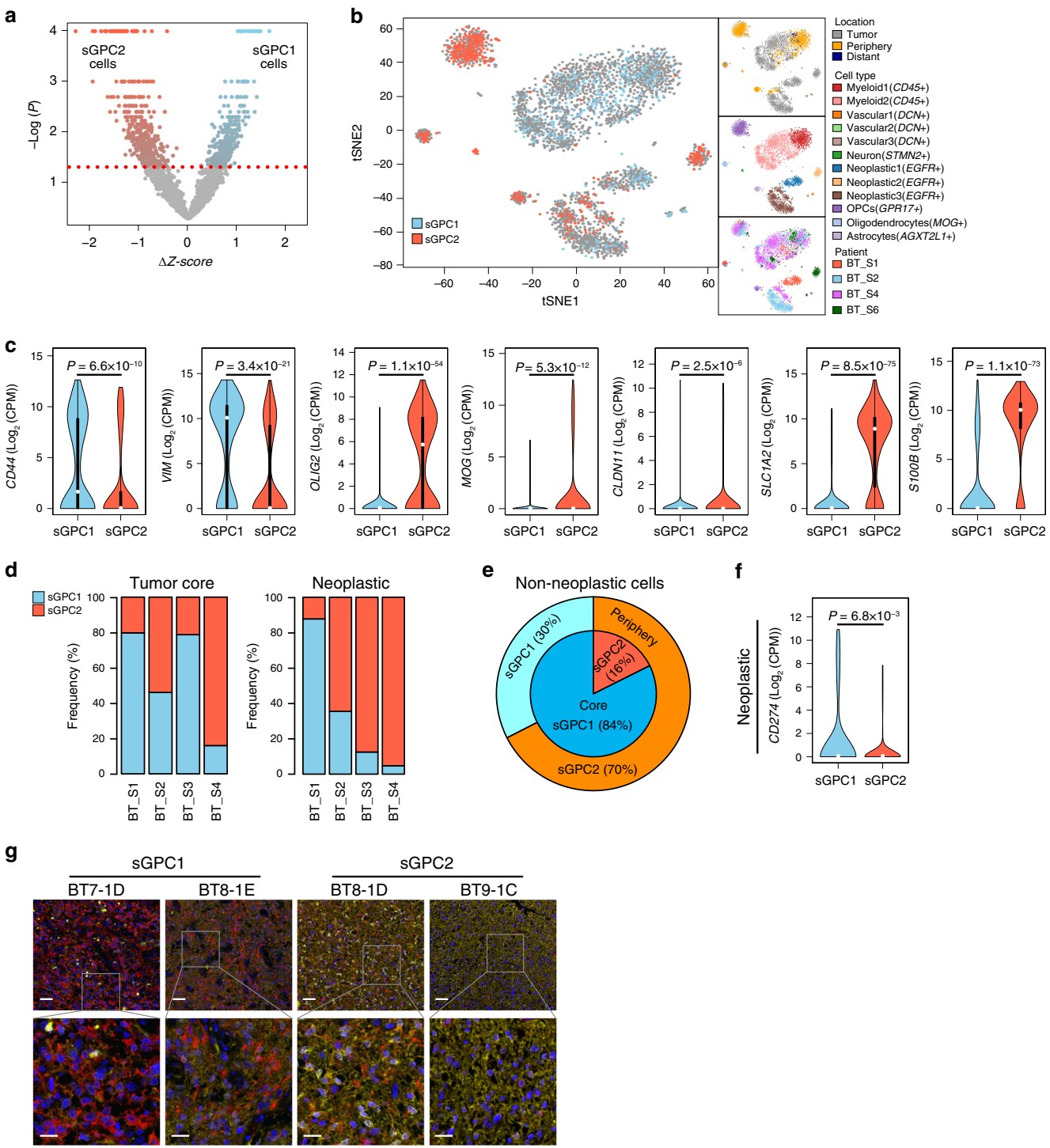

**Fig. 5 Classification of GBM single cells reveals tumor-layer-dependent and cell-type-dependent characteristics of GPC-subtypes. a** Classification of GBM single cells from the data set reported in Darmanis et al. [31]. The red dotted line indicates the statistical threshold used to determine the surrogate GPC (sGPC) subtype of single cells. **b** 2D t-SNE projection of single cells of surrogate-proteomic subtypes. t-SNE coordinates, single-cell annotations describing the location within the 3D tumor mass, cell type, and the patient origin (shown in right side panels) were obtained from Darmanis et al.[31]. **c** Subtype-specific mRNA expression of neural stem cell (NSC), oligodendrocyte, and astrocyte markers in single cells. CD44/VIM, OLIG2/MOG/CLDN11, and SLC1A2/S100B represent NSC, oligodendrocyte, and astrocyte markers, respectively. The violin plots represent density distributions. The description of box and whisker plots in violin is the same as in Fig. 3b. *P*-values were obtained by two-sided unpaired Wilcoxon rank-sum test. **d** Proportion of tumor core single cells (left) and neoplastic single cells (right) of two subtypes in four specimens in the data set generated by Darmanis et al. [31]. **e** Proportion of non-neoplastic single cells of two subtypes in the Darmanis et al. [31] data set. **f** Subtype-specific mRNA expression of an immune checkpoint ligand *CD274* (PD-L1) in neoplastic single cells. The description of the violin plot is the same as in **c**. *P*-values were obtained by two-sided unpaired Wilcoxon rank-sum test. **g** Representative images (*N* = 1) of multiplex fluorescent immunohistochemistry analysis of SMC-TMA samples using PHGDH (yellow) and Nestin (red) antibodies. Nuclei were stained with DAPI (blue). Sample IDs and their respective sGPC-subtypes are indicated above the image. Scale bars: 50 μm (upper panels), 20 μm (lower panels). Source data are provided as a Source Data file.

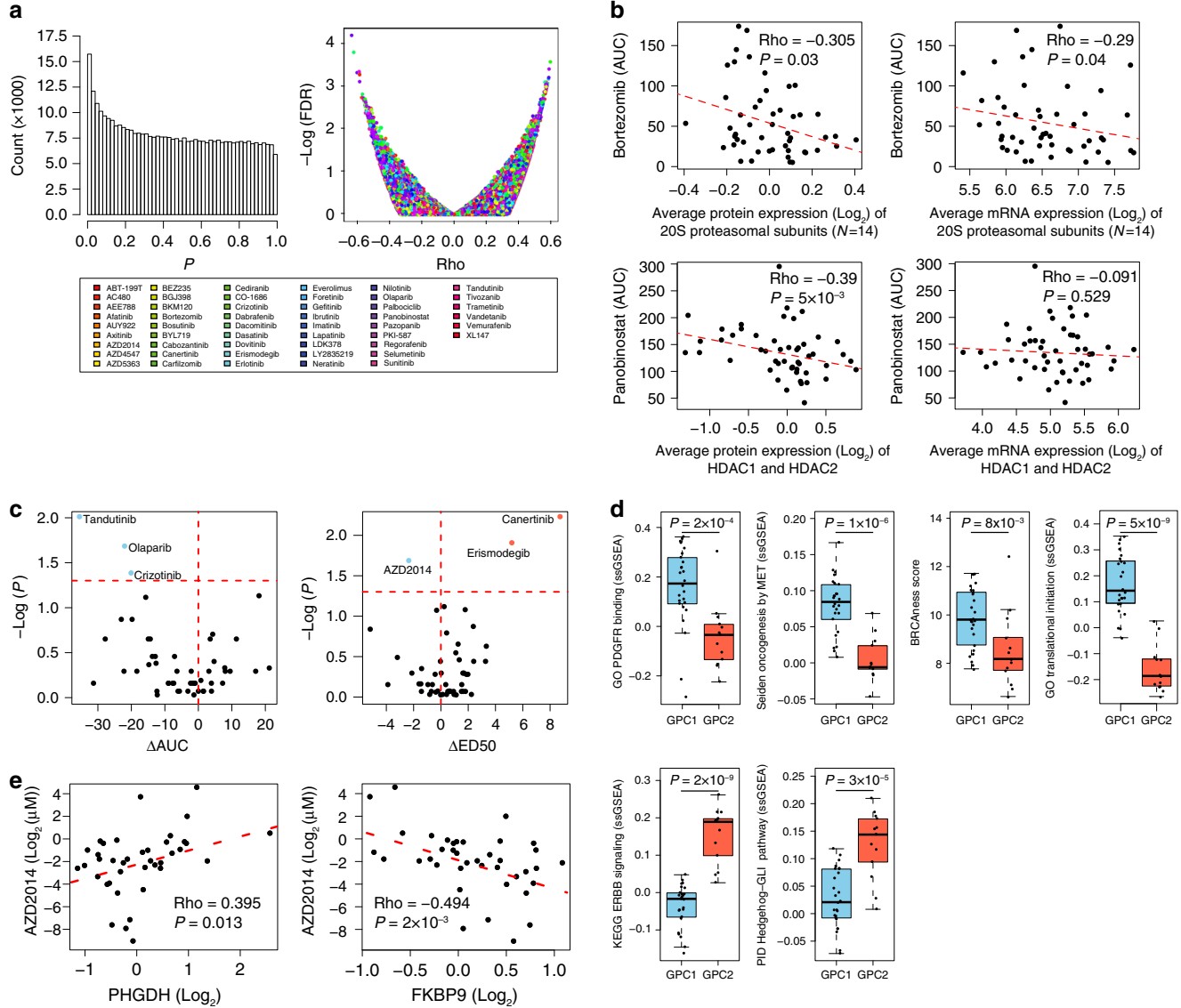

**Fig. 6 Subtype-specific and biomarker-dependent sensitivities to targeted therapies. a** Correlation tests between the drug–response (AUC and ED50) and the abundance of global proteins. (Left) Histogram of *P*-values, and (right) FDR score distribution. Rho and *P*-values were obtained by Spearman's correlation test. **b** Correlations between expression levels of the indicated proteins (or mRNAs) and responses to the indicated pharmacological compounds targeting them. Rho and *P*-values were obtained by Spearman's correlation test. **c** GPC1 or GPC2-selective cytotoxicity profile of the indicated targeted therapies to patient-derived tumor cells matched to the proteomic cohort. The x axes represent differences in mean AUC (left panel) and mean ED50 (right panel) values between GPC1 and GPC2 PDCs. Statistically significant GPC1 and GPC2-selective drugs were identified by two-sided Kolmogorov–Smirnov test (*P*-values < 0.05; horizontal dashed line) (blue: GPC1 sensitivity, red: GPC2 sensitivity). **d** Subtype-specific differences in relevant gene sets to GPC1-selective (upper panels) and GPC2-selective (lower panels) drugs in **c**. *P*-values were obtained with two-sided unpaired Wilcoxon rank-sum test. The number of samples for GPC1 and GPC2 is 26 and 13, respectively. The box-and-whisker plots represent the medians (middle line), first quartiles (lower bound line), third quartiles (upper bound line) and the ±1.5× interquartile ranges (whisker lines); the raw data are overlaid. See "Methods" for details. PDGFR platelet-derived growth factor receptor. **e** Correlation between the expression levels of the indicated proteins and the response to the indicated pharmacological compound. Rho and *P*-value were obtained by Spearman's correlation test. Source data are provided as a Source Data file.

As expected, hypersensitivities associated with the phosphorylation-mediated activation of a target protein were observed for several targeted agents. For example, hypersensitivities to afatinib, an EGFR inhibitor, correlated with EGFR-pY1197 (Supplementary Fig. 6c; left panel). Similarly, SRC pS17 and pY419 were associated with sensitivity to bosutinib (Supplementary Fig. 6c; middle and right panels).

For most drugs, however, stronger correlations were found for unrelated phosphoproteins or the non-activating phosphorylation sites of the target protein than for activating phosphorylation sites

in target proteins. For example, lapatinib, a dual EGFR and HER2 inhibitor showed marked association with EGFR-pT693, a known marker of receptor internalization (Supplementary Fig. 6d, upper left panel). Intriguingly, three other phosphoproteins commonly involved in RTK endocytic recycling—RAB4B-pS193, SNAP91-pT309 and ANK2-pS2516—were even more highly correlated with lapatinib sensitivity (Supplementary Fig. 6d), which suggests a possibility that EGFR recycling activity, rather than EGFR kinase activity, may determine responses to lapatinib. Other correlations that exhibited a high degree of

significance (FDR < 1%) included correlations between bosutinib and TSC2-pS1420, and between AZD4547 (FGFR inhibitor) and TOMM22-pS45 (Supplementary Fig. 6e). The functional impact of these phosphorylation sites and their relationships to the targeted agents are largely unknown.

Our final investigation aimed to identify targeted agents selective to the most aggressive GBMs, characterized by wild-type *IDH*, low PHGDH expression, and high FKBP9 expression in our study (Fig. 4d). Here, we found that the most significant correlations with both PHGDH and FKBP9 were found for AZD2014 (Fig. 6e), suggesting that AZD2014 might be a promising targeted therapy for the aggressive *IDH* wild-type GBM subtype.

In summary, our integrated analysis of pharmaco-proteomic data in patient-matched samples revealed proteomic-subtype-associated and protein biomarker-dependent sensitivities to targeted agents, which might provide additional insights into patient stratification strategies for GBM therapy in the future.

## Discussion

This study employed a large-scale quantitative proteomic approach using LC–MS/MS technology to characterize the intertumoral heterogeneity of 39 *IDH* wild-type GBM tumors. We found that *IDH* wild-type GBM can be divided into two stable proteomic subtypes, GPC1 and GPC2, which are primarily characterized by low or high expression of OXPHOS-related proteins, respectively. GPC1-subtype tumors displayed distinct Warburg-like proteomic features, with increased LDHA, PKM2, and HK2 expression that facilitates lactate production and confers resistance to hypoxic stress in cancer cells[34,35]. The Warburg effect drives the biosynthesis of nucleotides, lipids, and proteins to support rapid cell proliferation, as well as the disruption of tissue architecture to facilitate tumor motility and immune-cell evasion in the tumor microenvironment[24]. NADPH is a key component in this process, thus the elevated PPP, one-carbon pathway, and IDH1 levels in GPC1 tumors provide additional support for our hypothesis. As we found no direct correlation between the RNA and protein levels of OXPHOS-related proteins (presumably due to their dependence on post-translational pro-teolysis and protein turnover to control abundance)[15,36,37], our proteome-based classification exhibited no significant correlative relationship with previously identified RNA-based subtypes. Therefore, the proteomic subtypes represent previously unrecognized *IDH* wild-type GBM subgroups. However, some of the samples, particularly in GPC2, exhibited heterogeneous expression patterns, compared to other samples in the subtype, suggesting that increased sample size may lead to additional subtype(s) distinct from the two major GPC subtypes.

The identification of prognostic biomarkers for *IDH* wild-type GBM remains a challenge. Indeed, among the previously reported 271 gene expression biomarkers that are prognostic for GBM[28], only three were validated in our *IDH* wild-type GBM cohort at the protein level. Of these, PHGDH (a rate-limiting enzyme of serine biosynthesis) was elevated in GPC2 tumors of *IDH* wild-type GBM and showed the strongest association with long-term survival. IDH mutant proteins produce the oncometabolite D-2-HG and predict a favorable prognosis in glioma[9]. Similarly, PHGDH is known to have a promiscuous function to generate D-2-HG[29]. Thus, further research may be needed to evaluate whether the canonical PHGDH function or its promiscuous function is associated with favorable prognosis in *IDH* wild-type GBM.

Understanding the mechanistic connections among GPC1 tumor-specific features is important because these tumors carry poor prognostic biomarkers. Nestin is expressed in adult NSCs that reside in the subventricular zone (SVZ) of the brain: it forms a heterodimer with Vimentin during mitosis, promotes the dis-assembly of intermediate filaments, and supports the survival and renewal of neural progenitor cells. A recent study showed that 56% of human GBM cases originate from SVZ-derived glioma stem cells (GSCs)[26]. Thus, we hypothesized that GPC1 subtype might originate from SVZ-derived GSCs. In further support of this hypothesis, we found that GPC1 tumors expressed higher protein levels of phosphorylated cortactin and the Arp2/3 complex. Phosphorylated cortactin activates the Arp2/3 complex to mediate a mechanism by which cancer cells might facilitate actin filamentation and branching while remodeling the extracellular matrix to gain increased motility and invasiveness[38]. GSC maintenance depends on hypoxia inducible factor (HIF) 1α[39] consistently, we found that GPC1 tumors had activated HIF pathway (Supplementary Data 3). HIF1α inhibits cell differentiation by activating signaling pathways driven by Notch, NANOG, TGFβ, and SOX2. Metabolically, HIF1α activates GLUT, PFK1, HK2, and LDHA, thus inducing aerobic glycolysis and leading to increased glucose uptake due to a low ATP yield. Increased glucose uptake and lactate secretion subsequently facilitate immune evasion through immune checkpoint ligands and metabolic competition[40]. HIF1α also activates the one-carbon (folate) pathway to protect cancer stem cells from increased oxidative stress by increasing NADPH (and glu-tathione) production[41]. This process is consistent with our find-ings that NADPH-producing IDH1, the PPP and one-carbon pathway were elevated in GPC1 tumors.

We also demonstrated GBM intratumoral heterogeneity in terms of proteome subtype, although one subtype generally dominated a tumor. This finding is consistent with the recent observation reported in the Ivy Glioblastoma Atlas (http://glioblastoma.alleninstitute.org), that the differences in tumor characteristics separated by anatomical location are as large as the differences observed between other tumors[42]. A subtype switch observed in our longitudinal samples might also highlight intra-tumoral heterogeneity as a major challenge to successful GBM treatment. Our data imply that the two GPC subtypes might need to be controlled simultaneously in GBM treatment to prevent recurrence and have a therapeutic effect.

Finally, we assessed the direct chemical liabilities linked to proteomic information using the previously generated data set with 50 matched PDCs against 60 targeted anticancer drugs. Here, we demonstrated concordant subtype-specific target path-way (BRCAness) activation with olaparib showing efficacy for GPC1 tumor-matched PDCs. Olaparib is under clinical trials with radiotherapy (for MGMT unmethylated patients) and radiotherapy-temozolomide (for MGMT methylated patients) in newly diagnosed GBM[43]. Our proteomic classification may guide further patient selection criteria for these efforts. We also demonstrated that AZD2014, a dual mTORC1 and mTORC2 inhibitor, exhibited strong cytotoxicity toward the most aggres-sive PHGDH-low and FKBP9-high *IDH* wild-type GBM-derived PDCs. Despite the failure of mTORC1 inhibitors in clinical trials, dual mTORC1/2 inhibitors are increasingly gaining traction in GBM translational research because preclinical evidence indicates that mTORC2 has divergent roles from mTORC1 in facilitating GBM growth, invasiveness, and GSC proliferation[44,45], and that mTORC2 activity is selectively enhanced in grade IV tumors[46]. Our data are in line with these observations and thus should motivate further clinical studies.

In conclusion, our integrated pharmaco-proteogenomic ana-lyses highlight the importance of using proteomic data to understand the connections between GBM cellular origin, onco-genic signaling and metabolic diversity, all of which shape distinct binary molecular states. Our data illuminate unique therapeutic vulnerabilities coupled to these binary molecular states and

biomarker proteins and suggest potentially effective therapeutic strategies for GBM.

## Methods

**Sample acquisition.** Tumor specimens and their corresponding clinical records were obtained from patients who underwent surgical resection at Samsung Medical Center (SMC) and provided informed consent. SMC cohort 1 ($N = 50$) was used for the proteomic analysis, SMC cohort 2 ($N = 106$) was used as an independent data set for validation using RNA-based surrogate proteome signatures, and SMC-TMA cohort ($N = 120$) was used as an independent data set for validation by multiplex fluorescent immunohistochemistry. Detailed information about the specimens is provided in Supplementary Data 1. This study was approved by the SMC Institutional Review Board (201004004 and 200504001).

**Whole-exome sequencing (WES) data analysis.** The paired-end reads from FASTQ files were aligned to the UCSC human reference genome assembly (hg19) using Burrows–Wheeler Aligner BWA (version 0.6.2). The duplicated reads were removed using Picard (version 1.73), and local realignment was then performed around known insertions and deletions (indels) using SAMtools (version 0.1.18) and the Genome Analysis Tool Kit (GATK version 2.5-2). dbSNP (version 135) was subsequently used for the realignment and recalibration process. The resulting bam files were used for mutation calling and copy number analysis.

SNVs were identified using MuTect (version 1.1.4) with the following criteria: a Phred score >15 and coverage >20 in both the tumor tissue and the matched normal tissue. Indels were identified by SomaticIndelDetector (GATK version 2.2) based on a mapping quality score >15 and a coverage >10 in both the tumor tissue and the matched normal tissue. The variants were annotated using dbSNP (version 135), 1000 Genomes Project (Phase I), and Exome Sequencing Project (ESP6500SI-V2). The resulting variants were annotated using the Variant Effect Predictor (VEP version 37.75)[47]. After removing synonymous variants, the nonsynonymous and splicing variants were taken forward for further study. Hotspot mutations were annotated using databases downloaded from www.cancerhotspots.org (Hotspot Results V1) and www.3dhotspots.org (3D Hotspot Results).

For the copy number analyses, BAM files from WES of 50 GBM and 24 matched normal blood samples were used to generate gene-based read-count matrices using Bedtools, according to the Ensembl gene table (version 37.75). Subsequently, 1 was added to the read counts to prevent negative infinity values for the $\log_2$ transformation. Genes with mean read counts <20 were filtered out. The read counts were normalized to counts per million (CPMs) using the edgeR package (version 3.20.9). The normalized values of the tumors were divided by that of the matched normal samples to calculate the $\log_2$-ratio. A circular binary segmentation algorithm was implemented with the $\log_2$-ratio values using the DNAcopy package (version 1.52.0). Genomic identification of significant targets in cancer (GISTIC) 2.0 using the default parameters was applied to the segmented data to identify regions that were frequently altered in DNA copy number.

**RNA-sequencing data analysis.** GSNAP (version 2012-12-20)[48] was used to align the reads of SMC cohort 1 to the reference genome GRCh37, and STAR (version 2.5.4b) aligner[49] was applied to SMC cohort 2. The normalized gene expression values in fragments per kilobase of exon per million fragments mapped (FPKMs) were quantified by Cufflinks (version 2.0.2). Finally, the (FPKM + 1) values were transformed to the $\log_2$ scale.

To determine the RNA subtype, single-sample gene set enrichment analysis (ssGSEA)[50] was applied to the $Z$-score normalized expression data to calculate enrichment scores (ESs) for RNA subtypes as defined by Verhaak et al.[8]. The subtype with the highest ES was used as the representative subtype for each sample.

**Materials for quantitative proteomic analysis.** A sequencing-grade modified trypsin/LysC mix was purchased from Promega (Madison, WI, USA), and Tris (2-carboxyethyl)phosphine hydrochloride, tandem mass tag (TMT) isobaric reagents and a Pierce graphite spin column were purchased from Thermo Fisher Scientific (Waltham, MA, USA). Water and organic solvents were obtained from J.T. Baker (Center Valley, PA, USA). Titansphere™ Phos-TiO (10-μm bulk) was purchased from GL Science (Tokyo, Japan).

**Protein extraction, digestion, and TMT peptide labeling.** Tumor and adjacent normal tissue samples were carefully washed in PBS buffer on ice to remove the blood and then individually cryo-pulverized using a cryoPREP device (CP02, Covaris). Each tissue specimen (32–243 mg in total wet tissue weight) was placed in a cryovial (Covaris, 430487) on dry ice, transferred to a Covaris tissue bag (TT1, Covaris), placed into liquid nitrogen for 30 s and then pulverized at impact level 3. The tissue powder from each tissue was then placed in a sonication tube (Covaris, 002109) and mixed with lysis buffer [8 M urea, 0.1 M $NH_5CO_3$, 1 mM PMSF and 1× phosphatase inhibitor cocktail (ThermoScientific, Pittsburg, PA, USA)]. The lysis buffer volume varied depending on the total tissue weight (ca. 1 mL for 20 mg). Tissue lysis was performed by sonication using a focused ultrasonicator (Covaris, S220) at a setting of 2 W (intensity of 5) for 5 s followed by 36 W (intensity of 10) for 20 s and 0 W (intensity of 0) for 10 s. The sonication cycle was

repeated 20 times at 16 °C. The homogenate was centrifuged at 16,000×g and 20 °C for 10 min (5810R, Eppendorf), and the supernatant was transferred to a new tube. The protein concentration was determined by bicinchoninic acid protein assay (ThermoFisher Scientific, Waltham, MA, USA).

The proteins (700 μg) were subjected to disulfide reduction with 5 mM Tris(2-carboxyethyl)phosphine at room temperature for 2 h and alkylation with 15 mM iodoacetamide at room temperature for 1 h in the dark. Subsequently, the samples were diluted 10-fold with 0.1 M $NH_5CO_3$ to reduce the concentration of urea to 0.8 M. The protein sample was digested overnight at 37 °C using a Trypsin/LysC protease mixture at a 1:25 enzyme–substrate ratio. The digested samples were cooled at room temperature, and the digestion was quenched by acidification with trifluoroacetic acid (TFA) at a final concentration of 0.5%. The sample was subsequently purified/desalted through HLB solid-phase extraction (SPE) (Sep-Pak, Waters), dried in vacuo and stored at −20 °C until further use.

The dried peptides were resuspended in labeling buffer (0.1 M Triethylammonium bicarbonate buffer, Sigma Aldrich), and the peptide concentration was determined using a Nanodrop spectrophotometer (ThermoFisher Scientific) at 280 nm wavelength. An aliquot equivalent to 500 μg of each sample was immediately labeled with 4 mg of each TMT channel, except for TMT 126, which was prepared according to the manufacturer's instructions. For the first batch (sets 1–6), samples from sets 1 through 6 were combined to obtain a global internal standard 1 (GIS1); this standard was labeled with TMT channel 126. GIS2 was used for the second batch (sets 7–11), which was obtained by pooling sets 7–11. Following incubation at room temperature for 1 h, the reaction was quenched with hydroxylamine at a final concentration of 0.3% (v/v). The TMT-labeled samples were pooled at a 1:1:1:1:1 ratio. The sample arrangement is shown in Supplementary Data 1. The combined sample was subsequently purified/desalted using HLB-SPE, dried in vacuo, and stored at −20 °C until further use. The combined sample comprising the first batch was dried in vacuo and subsequently desalted using HLB-SPE. By contrast, the combined sample for the second batch was directly desalted and subsequently dried in vacuo because drying samples in the presence of hydroxylamine is detrimental to phosphopeptides[51].

**Peptide fractionation and preparation of proteome samples.** The TMT-labeled peptides were fractionated by bRPLC using an Agilent 1290 Infinity LC System (Agilent Technologies). Chromatography was performed with an XBridge BEH130 C18 column (4.6-μm i.d. × 250-mm length; pore size of 130 Å and particle size of 3.5 μm; Waters Corporation, Milford, MA, USA) at a flow rate of 0.5 mL/min. The mobile phases were 10 mM $NH_4HCO_2$ (pH 10) as phase A and 10 mM $NH_4HCO_2$ (pH 10) in 90% ACN (pH 10) as phase B. The peptides were dissolved in 110 μL mobile phase A and then injected into a 100-μL sample loop. The gradient was 2–5% B for 10 min, 5–40% B for 40 min, 40–70% B for 15 min, 70% B for 10 min, and 70-5% B for 15 min. Fractionation was performed by collecting 84 tubes (0.8 min/tube) throughout the chromatographic run. Eighty-four fractions were pooled to obtain 12 concatenated fractions based on the following rule: a set of an arithmetic sequence with a common difference of 12 was pooled into one concatenated fraction; for instance, fractions with numbers 1, 13, 25, 37, 49, 61, and 73 were pooled to generate concatenated faction 1. A total of 5% of the volume of each fraction was allocated to global proteome analysis and dried. The remaining 95% of the concatenated fractions were further combined into 12 fractions, and the flow-through fractions from bRPLC were also combined into one fraction for phospho-peptide enrichment and dried. For all experiments, the phosphopeptides were subjected to metal oxide affinity chromatography using titanium dioxide beads (10 μm, Titansphere Phos-TiO Bulk)[52,53]. The dried peptide and $TiO_2$ beads were preincubated separately in a solution of 3.45 M lactic acid (302 mg/mL), 60% ACN and 0.3% TFA (one fraction of peptide in 100 μL of the solution; 2 mg of beads in 10 μL of the solution). The two preincubated mixtures were combined and further incubated for 30 min at 25 °C with agitation. After incubation, the beads enriched with phosphopeptides were collected by centrifugation, and the unbound super-natant from the three fractions was pooled into one fraction for double $TiO_2$ enrichment[54]. The beads were washed with 1% TFA in 30% ACN and loaded onto a C8-plugged tip (Diatech Korea, Seoul, Korea). The bound phosphopeptides were eluted with 1.5% $NH_4OH$ and then with 5% pyrrolidine in a single tube. The eluates were directly acidified with 1% TFA and desalted using graphite spin col-umns (ThermoScientific) according to the manufacturer's instructions. The phosphopeptides were dried and resuspended in 0.4% acetic acid.

**Liquid chromatography and tandem mass spectrometry.** The dried peptide samples were reconstituted in 0.4% acetic acid, and an aliquot containing ~1 μg of the sample was injected from a cooled (10 °C) autosampler into a reversed-phase Magic C18aq (Michrom BioResources, Auburn, CA, USA) column (20 cm × 75 μm, packed in-house) on an Eksigent nanoLC-ultra 1D plus system at a flow rate of 300 nL/min. Before use, the column was equilibrated with 90% buffer A (0.1% formic acid in water) and 10% buffer B (0.1% formic acid in acetonitrile). The peptides were eluted with a linear gradient from 5 to 40% buffer B over 100 min and 40 to 80% buffer B over 5 min and then subjected to an organic wash and aqueous re-equilibration at a flow rate of 300 nL/min with a total run time of 130 min. The HPLC system was coupled to a Q-Exactive mass spectrometer (ThermoFisher Scientific, Bremen, Germany) operated in data-dependent acqui-sition mode. Survey full-scan MS spectra ($m/z$ 400–2000) were acquired at a

resolution of 70,000. The source ionization parameters were as follows: spray voltage, 2.5 kV; capillary temperature, 300 °C; and s-lens level, 44.0. The MS/MS spectra of the 12 most intense ions from the MS1 scan with a charge state of 1–5 were acquired with a fixed first $m/z$ of 120 along with the following options: resolution, 17,500; automatic gain control target, 1E5; isolation width, 2.0 $m/z$; normalized collision energy, 27%; dynamic exclusion duration, 90 s; and ion selection threshold, 4.00E + 03 counts.

**Peptide and protein identification and quantification.** Peptide and protein identification and quantification were performed using MaxQuant[55] 1.5.6.0. The mass spectrometry raw files were searched against the Swiss-Prot human database (released in March 2014; http://www.uniprot.org) using the Andromeda search engine included in MaxQuant. The following MaxQuant search parameters were used: semispecific trypsin was selected as the enzyme; the carbamidomethylation of cysteine was set as a fixed modification; N-terminal protein acetylation and oxidation (M) were set as variable modifications; and phosphorylation (STY) was set as a variable modification for phosphorylation-enriched samples. The reporter ion was set as six-plex TMT for quantification. Peptide matches were filtered by a minimum length of eight amino acids and no miscleavages were allowed. The false discovery rate (FDR) was set to 0.01 at both the protein and peptide spectrum match (PSM) levels. Proteins identified by at least two unique peptides were used. For protein quantification, the minimum ratio count was set to two, and the peptide for protein quantification was set as unique. Other settings were kept at their default values. In total, 9367 protein groups, 179,234 stripped peptides, and 2,750,407 peptide spectral matches (PSMs) were identified from the global proteome. In the case of the phosphoproteome, 8019 phosphorylation sites, 16,377 phosphorylated peptides, and 276,153 PSMs were identified. The mass spectrometry proteomics data have been deposited to the ProteomeXchange Consortium (http://proteomecentral.proteomexchange.org) via the PRIDE partner repository with the data set identifier PXD015545. The protein intensity of a sample was divided by the corresponding global internal standard (GIS) and converted to the $\log_2$ scale. The normalized $\log_2$ intensities were then sample-wise median centered across all proteins before protein-wise median centering across samples belonging to a GIS batch. Finally, the normalized abundance values obtained from two GIS batches (30 samples from batch 1, 24 samples from batch 2) were combined. Samples with no intensity values were arbitrarily given the second smallest value of all samples to avoid negative infinity. A total of 8034 proteins quantified in all GISs were used for analysis.

To convert the abundances of the phosphopeptides to that of phospho-sites, the mass spectral intensity values from all the phosphopeptides containing a particular phospho-site ($s$) were averaged at the levels of mono-, di-, or tri- or more phosphorylated peptide signal intensities, respectively. These three levels of intensity values ($I_{sn}$) for a single phospho-site were subjected to the following equation:

$$I_s = \sum_{n=1}^{3} \frac{I_{sn}^2}{\sum_{n=1}^{3} I_{sn}}, \tag{1}$$

where $I_s$ represents a weighted squared sum of the intensity value for a phospho-site ($s$). This weighting was performed to give more weight to the phosphorylation class providing higher intensities, and the $I_s$ values were then further log2 transformed. For the samples with no intensity value for a phospho-site, the second minimum value of all the samples was arbitrarily set to avoid negative infinity. Due to the relatively high sparsity of phosphorylation data, phosphopeptides ($N = 11,346$) quantified in at least three GISs (approximately ten samples or more) were considered for the quantification of the phospho-site level.

Based on the observation that the global and phosphoprotein abundance data of the six subgroups (normal, tumor, GPC1, GPC2, *IDH* wild-type, *IDH* mutant, low-grade glioma (LGG), and GBM) largely follow a Gaussian distribution, as determined by the Shapiro–Wilk test, Student's *t*-test was used to identify differentially expressed proteins (DEPs) and differentially expressed phosphoproteins (DEPPs) from the four comparisons (tumor vs. normal, GPC1 vs. GPC2, GBM vs. LGG, and *IDH* wild-type vs. mutant). The DEPs and DEPPs whose *P*-values were <0.05 with their corresponding FDR scores were selected for further analyses. The DEPPs were further filtered when these were found as DEPs in the same direction in the comparison.

**Variant peptide identification.** To determine the overlaps between the variants called from the WES data and those identified from the mass spectrometry data, variant peptides were identified using a multistage search approach[56] and a unified protein database. A unified protein database consists of both a reference and a sample-specific protein database. The Swiss-Prot human protein database (version 2014/03) was used as the reference protein database. A sample-specific protein database was constructed using the following four types of information: sample-specific protein expression, sample-specific genomic variations, fusion gene prediction, and common contaminants. For this database, the transcript models in Ensembl 75 (released in February 2014) whose FPKM values were >1 were used, similar to our previous study[57]. The sample-specific variant peptide database was built by applying SNVs and indels called from the WES data and observed in the RNA-seq data to the transcript models. The resulting RNA sequences were then

converted into amino acid sequences allowing up to three missed cleavages on both sides, in accordance with the methodology used in our previous study[57]. Sometimes, SNVs and indels can result in stop gain or stop loss. The variants resulting in stop gains were ignored because it was impossible to distinguish the variant peptides whose translations were stopped by missed cleavage parameters or a novel stop codon. Conversely, if a stop loss occurred, the sequences were translated if (1) up to 20 amino acids whose read depths at the translated positions were >3 or (2) until a new stop codon was found. Due to the constraint on the read depth, short peptides could be generated due to early termination. To prevent short peptide generation, up to five missed cleavages in the N-terminus direction were allowed during translation. For the sample-specific fusion gene database, fusion genes were predicted from the RNA-seq data and translated from the fusion junction in both directions based on the frame of the upstream gene.

To identify variant peptides, the MS/MS spectra were searched using a multistage approach[56], which consisted of the sequential application of MS-GF+[58] with a unified database and the in-house software MODplus with the unified database over the set of spectra that were not identified during the first stage. The parameters used for the MS-GF+ search were as follows: precursor error tolerance, 10 ppm; isotope error range, −1 to 2; fragmentation method, HCD; and instrument, Q-Exactive; variable posttranslational modification (PTM), M-oxidation; and two fixed modifications, C-carbamidomethyl and K/N-term-TMT. The second-stage MODplus search focused on the identification of a variety of PTMs. The MODplus search parameters were the following: precursor error tolerance, 10 ppm; fragment ion tolerance, 0.025 Da; isotope error range −1 to 2; instrument, QTOF (equivalent to Q-Exactive with HCD); 46 variable PTMs (this parameter allows identification of modified peptides with multiple modifications within a range of −480 to 470 Da; a list of the variable modifications is provided in Supplementary Data 2); and two fixed modifications, C-carbamidomethyl and K/N-term-TMT. Furthermore, MODplus identified modified peptides with multiple modifications within a given mass range, which is a user-specific parameter. All search steps were executed with the trypsin enzyme, TMT protocol, and semi-tryptic search. In each step, peptides <8 amino acids and an estimated 1% FDR were discarded using a target-decoy approach. The MODplus search considered De-TMT modifications that discriminated TMT-free PSMs. The TMT-free PSMs were discarded because it was unable to determine in which sample the peptide was expressed. The overlapping somatic variants with WES data were identified based on their genomic coordinates. Finally, the Jaccard coefficient was calculated.

**Protein isoform analysis.** In total, 18 protein isoforms (sharing a gene symbol) were quantified across the GISs and had different quantification values; these corresponded to nine genes (*CAPZB*, *EPB41L3*, *IKBIP*, *MAP2*, *MAP4*, *PFN2*, *PKM*, *RTN1*, and *SNX32*). Among the 18 isoform groups, only PKM isoforms exhibited mutually exclusive expression patterns in two GPC subtypes: PKM1 was significantly elevated in GPC2, whereas PKM2 was in GPC1. For the comparison of PKM isoforms, we selected peptides belong either to PKM1-specific exon 9 or PKM2-specific exon 10. The peptide intensities were normalized based on the corresponding GISs and transformed to the $\log_2$ scale. The average peptide expression values were then calculated according to the specific sequences for either PKM1 or PKM2. Five *PKM1* isoforms (ENST00000319622, ENST00000389093, ENST00000565154, ENST00000565184, and ENST00000568459) and one *PKM2* isoform (ENST00000335181) were used. The average expression of the isoforms was used for PKM1.

**Calculation of RNA to protein correlation.** The Ensembl gene IDs and UniProt IDs were converted into an official gene symbol using the BioMart (version 2.34.2), org.Hs.eg.db (version 3.5), EnsDb.Hsapiens.v75 (version 2.99), UniProt.ws (version 2.18), and rentrez (version 1.2.1) packages. The Spearman correlation coefficients and *P*-values for 4071 genes were calculated using the R function cor.test.

**Identification of proteomic subtypes.** Multiple protein IDs with identical numeric values across all samples, which likely indicate isoforms, were grouped into one ID to prevent undesirable effects from redundant protein IDs. Subsequently, a consensus clustering algorithm was applied to the protein expression matrix with default parameters and 1000 iterations. The optimal number of clusters was determined by assuming that the K value represents the minimal PAC score[59].

**Gene set enrichment analyses (GSEA) and network analysis.** Gene set analysis (GSA) was conducted with official symbols from the top 10% PC1 high loadings and DEPs/DEPPs. A total of 9404 unique pre-annotated gene sets obtained from the CORUM (version 3.0), C2, and C5 MSigDB were used after removing redundant, too-small (<5), and too-large (>150) gene sets. Gene sets with at least four hits were selected from the given symbol lists. The gene symbols corresponding to all global proteins or all global and phosphoproteins were used as the background for PC1-GSA and DEPs/DEPPs-GSA, respectively. The statistical significance of the gene set enrichment was evaluated based on the hypergeometric test, followed by multiple testing correction with the false discovery rate (FDR). Gene sets with FDR < 1% were selected for normal/tumor, whereas those with FDR < 10% were selected for PC1-loadings to examine the relationships between gene

sets using the EnrichmentMap algorithm based on an overlap coefficient cutoff of 0.5[60]. For the DEPs and DEPPs, the signaling network, enzyme–substrate interactions, and transcription factor–target interactions were further analyzed using the literature-curated OmniPath database[61]. BRCAness score was estimated at the mRNA level using the methods described by Konstantinopoulos et al.[62].

**Processing of public data**. Gene expression and metadata of TCGA were downloaded using the TCGAbiolinks package (version 2.6.12)[63]. After 1 was added to each gene expression value from the TCGA data, the values were transformed to the $\log_2$ scale.

**Surrogate GBM proteomic cluster (sGPC) subtyping**. To extend our analysis to larger cohorts lacking proteome data, we constructed a random forest model using the differentially expressed genes (DEGs) between GPC1 and GPC2 samples and predicted sGPC subtypes of IDH-WT GBM tumor samples in a separate independent cohort ($N = 106$, SMC cohort 2) and TCGA cohort ($N = 149$). DEGs between two GPC subtypes in IDH wild-type GBMs of SMC cohort 1 were obtained by Student's t-test, and the top 100 DEGs were selected based on the P-values. The individual RNA expression data of the other cohort were then merged with the corresponding SMC cohort 1 data, and the merged matrix was subjected to quantile normalization to reduce the batch effects. The random forest model of the normalized SMC cohort 1 was trained using the randomForest package (version 4.6-14), and the optimal parameters were selected using the caret package (version 6.0-8) with 1000 iterations of fivefold cross-validation. The model was applied to the normalized SMC cohort 2 and samples with a GPC subtyping probability of at least 60% were used for downstream analyses.

To address the sparsity issues in the Darmanis single-cell data[31], DEGs were identified from the 1000 genes with the highest expression in the SMC 1 cohort and those that were expressed in at least 50% of all single cells. To predict the sGPC subtype of each single cell, an equal number of DEGs (FDR < 10%) were selected from both sides. Likewise, an equal number of DEGs (100 GPC1-high and 100 GPC2-high) from the CCLE, Yonsei, and ANOCEF cohort data were selected according to Student's t-test P-values. The selected gene expression values were converted into Z scores. $\Delta Z$ score was calculated using the following equation:

$$\Delta Z \text{ score} = \text{mean}(\mathbf{Z}_{\text{GPC1}}) - \text{mean}(\mathbf{Z}_{\text{GPC2}}), \qquad (2)$$

where, $\mathbf{Z}_{\text{GPC1}}$ and $\mathbf{Z}_{\text{GPC2}}$ represent vector of Z scores of either GPC1- or GPC2-high genes. The P-value for the $\Delta Z$ score was estimated by 1000X permutations of the gene labels. The sGPC subtype was determined if a single cell had permutation P-values < 0.05.

**Survival analysis**. Multi-samples with different GPC subtypes were excluded to prevent the effects of multiple samples on the survival analysis. If samples shared a GPC subtype, they were considered a single sample. The survival analysis was performed using the survival package in R (version 2.42-6). 271 previously reported gene expression-based prognostic markers[28] were validated in our SMC1 cohort by univariate Cox regression analysis of survival rate under the proportional hazards assumption using the coxph function in "survival" package (ver 2.43-3) of R. If there are multiple samples per patient, a mean expression value was used for a protein.

**Multiplex fluorescent immunohistochemistry**. The tissue microarray (SMC-TMA) consisted of 120 tissue samples that were formalin fixed, paraffin embedded (FFPE), and sectioned (2-mm thickness): 6 normal samples, 35 low-grade gliomas, 1 IDH-mutant GBM, 3 IDH-status unknown GBMs, and 75 IDH-wild-type GBMs, including 14 SMC2 tumors. The FFPE tissues on slides were deparaffinized and rehydrated for multiplex immunohistochemistry staining. Epitope retrieval was performed using BOND Epitope Retrieval Solution 2 kits (Leica Biosystems, AR9640). Immunofluorescent signals were visualized using the OPAL 7-Color automation IHC kit (Akoya, NEL82100KT), TSA dyes 570 (PHGDH; Atlas Antibodies, RRID: AB_1855299, 1/1000), 690 (Nestin; Atlas Antibodies, RRID: AB_1854381, 1/700), and spectral DAPI. The stained slides were coverslipped using HIGHDEF® IHC fluoromount (Enzo, ADI-950-260-0025) and scanned using a Vectra® 3.0 Automated Quantitative Pathology Imaging System (PerkinElmer). Color separation, cell segmentation, and cell phenotyping were performed on inForm Advanced Image Analysis software (version 2.2, PerkinElmer) to extract image data. PHGDH and Nestin-positive cells were determined by thresholds of 0.6 (PHGDH) and 1.25 (Nestin), respectively.

**Cell lines**. SNU466, SNU201, SNU626, A172, HS683, SNU1105, and T98G were purchased from Korean Cell Line Bank. KNS81 cells were obtained from the JCRB cell bank. U87MG and U87MG-IDH1-R132H cells were purchased from ATCC. Cells were grown in RPMI-1640 medium (Gibco, 11875-093) supplemented with 10% fetal bovine serum (Gibco, 16000-044) and 1% penicillin–streptomycin (Gibco, 15140122). The absence of mycoplasma contamination was confirmed in all cell lines by e-Myco VALiD Mycoplasma PCR detection kits (LiliF, 25299). Short tandem repeat profiles of the GBM cell lines are provided in Supplementary Data 5.

**Western blot and antibodies**. Cells were lysed in RIPA buffer (Sigma-Aldrich, R0278) with protease inhibitor cocktail (Genedepot, P3100) and phosphatase inhibitor cocktail (Genedepot, P3200). Protein concentrations were determined by the Bradford protein assay (BIORAD, 500-0006), and equal amounts of protein were loaded and separated in sodium dodecyl sulfate-polyacrylamide gels (SDS-PAGE). Proteins were then transferred to nitrocellulose membranes (BIORAD, 1620177). After blocking with 5% skim milk, the membranes were probed with primary antibodies. Antibodies used in this study were as follows: STAT1 (Cell Signaling Technology, AB_2799965, 1/1000); pSTAT1 Serine-727 (Cell Signaling Technology, AB_2773718, 1/1000); PHGDH (Cell Signaling Technology, AB_2737030, 1/1000); Nestin (Abcam, AB_10859398, 1/1000); FKBP9 (Novus Biologicals, AB_11005959, 1/1000); β-actin (Cell Signaling Technology, AB_2223172, 1/1000); IDH1-R132H (DIANOVA, AB_2335716, 1/1000); IDH1 (Cell Signaling Technology, AB_10950504, 1/1000); Anti-rabbit IgG (Jackson ImmunoResearch, AB_2307391, 1/5000); and Anti-mouse IgG (Jackson ImmunoResearch, AB_2307392, 1/5000). After washing three times, the membranes were incubated with secondary antibodies. Band signals were developed with ECL western blotting substrate kit (Pierce, 32106).

**cDNA Transfection**. The pCMV6-GFP-PHGDH (RG203949) was purchased from OriGene. A172 and SNU201 cells were seeded at 300,000 cells per well in 6-well plates. After overnight incubation at 37 °C, cells were transiently transfected with 2 μg of PHGDH cDNA or an empty vector using Lipofectamine 2000 (Invitrogen, 11668019) according to the manufacturer's instructions. After 24 h, PHGDH expressing cells were collected by trypsinization and resuspended in spheroid forming ECM solution (R&D Systems, 3500-096-K).

**3D spheroid cell invasion assay**. 3D invasion assay was performed using Cultrex 3D Spheroid Cell Invasion Assay (R&D Systems, 3500-096-K) according to the manufacturer's instructions. Briefly, resuspended cells in 1× spheroid forming ECM solution were seeded at 4000 cells per well in 96-well plates and incubated at 37 °C for 48 h for spheroid formation. Once the spheroids were formed, they were embedded in an invasion matrix and supplemented with culture medium containing NCT-502 (MedChemExpress, HY-117240) or DMSO. 3D spheroid invasion assay plates were incubated for 72 h. Invading spheroids were photographed using an inverted phase-contrast microscope (Olympus, IX73) with CellSence standard 1.15 software (Olympus) at ×4 magnification. For quantification of spheroid invasion, distances of invading cells were measured by ImageJ software. For this, the three longest protrusions or migrated single cells from each of the quadrants were combined, and the median value of the 12 distances per condition were used to compare the invasiveness of the spheroids.

**Phosphoglycerate dehydrogenase (PHGDH) activity assay**. Basal PHGDH activities of the nine GBM cell lines were measured by PHGDH activity assay kits (BioVision, K569) according to the manufacturer's instructions. Briefly, $10^6$ cells were lysed and centrifuged at $10,000 \times g$ for 5 min at 4 °C. The supernatant was mixed with saturated 4.32 M ammonium sulfate (BioVision, 7096) to remove interferences. The mixture was then placed on ice for 30 min and centrifuged at $10,000 \times g$ for 10 min. The pellet was resuspended in the assay buffer. PHGDH induced changes in the probe signals were measured by EnVision 2105 (PerkinElmer) at OD 450 nm. HS683 and SNU1105 cells were treated with NCT-502 (MedChemExpress, HY-117240) or DMSO for 2 h before measuring enzyme activity.

**Analysis of high-throughput drug screening data set**. Drug screening data[19] were used with some analytic modifications. The normalized data per drug per concentration were smoothed by removing data points with an abnormally high viability value (>1.5 and >3× interquartile range from the third quartile viability value of all PDCs) based on the assumption that the anticancer drugs in our panel are not expected to increase the proliferation of PDCs significantly. If a PDC showed no related data for any drug concentration, the pair was not used for further studies. Area under the curve (AUC) values were calculated using the trapezoidal method: a low AUC indicates high cell line sensitivity to the drug. ED50 values were calculated using the drc package (version 3.0-1) with a four-parameter log-logistic fit. If the imputed ED50 values were higher than the highest tested dose (20 μM), 20 μM was assigned as the ED50 of the drug. Statistically significant associations between phosphoproteins and drug responses across 50 PDCs were identified by subjecting all the phosphoproteins in Supplementary Data 2 to a Spearman's correlation test with the ED50 and AUC drug–response data.

**Reporting summary**. Further information on research design is available in the Nature Research Reporting Summary linked to this article.

## Data availability

The data supporting the findings of this manuscript are available from the corresponding author upon reasonable request. Previously published WES and RNA-Seq data that were reanalyzed here are available from EGA (https://www.ebi.ac.uk/ega/studies/EGAS00001002515). The mass spectrometry proteomics data have been deposited to the

ProteomeXchange Consortium via the PRIDE partner repository with the data set identifier PXD015545 (http://proteomecentral.proteomexchange.org). Normalized single-cell gene expression data (GBM_normalized_gene_counts.csv) was downloaded from http://gbmseq.org. CCLE's mutation (CCLE_DepMap_18Q1_maf_20180207.txt), gene expression (CCLE_RNAseq_genes_rpkm_20180929.gct.gz), and metabolome data (CCLE_metabolomics_20190502.csv) were downloaded from https://portals.broadinstitute.org/ccle/data. Gene expression microarray data for the Yonsei[64] and ANOCEF[65] cohorts were downloaded from GEO (https://www.ncbi.nlm.nih.gov/geo/query/acc.cgi?acc=GSE131837) and Arrayexpress (https://www.ebi.ac.uk/arrayexpress/experiments/E-TABM-898/), respectively. FPKM-normalized RNA-Seq and survival data for 149 IDH wild-type TCGA-GBM tumors were downloaded from https://portal.gdc.cancer.gov. The source data underlying Fig. 4g and Supplementary Figs. 1f and 4c, d, e are provided with this paper.

## Code availability

The code for generation of heatmaps and other graphics is available for download at https://github.com/hk-lab-software/gbm2020/. Source data are provided with this paper.

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

## Acknowledgements

We thank Sang Bum Kim and Daniel H. Kim for critical reading of the manuscript and their helpful discussions. The biospecimens for this study were provided by the Samsung Medical Center BioBank. This study was supported by grants from the National Research Foundation of Korea (2020R1A2C3007792, 2017M3A9F9030559, 2017M3C9A5031595, 2012M3A9B9036676, 2017R1E1A1A01077412, 2019R1A2C3004155, and 2019H1A2A1075632) and the Korea Health Technology R&D project through the Korea Health Industry Development Institute (HI14C1324, HI14C3418).

## Author contributions

Conceptualization, H.S.K. and C.L.; specimens and clinical data, D.-H.N., H.J.C., and J.K.; sample processing, S.J. and Y.K.; mass spectrometry, J.Y.; proteomic data analyses, S.O., H.K., H.L., M.J.O., and J.H.H., genomic data analyses, S.O., J.K.S., and H.J.C.; Mass spectrometric data analyses, J.Y., S.J., W.L., H.L., and S.C.; in vitro experiments and immunohistochemistry, J.-H.K., S.-J.Y., J.H.C., S.H.K., S.-G.K., E.C., and N.-G.H., supervision, H.S.K., C.L., D.-H.N., and E.P.; writing, H.S.K., S.O., and J.Y.

## Competing interests

The authors declare no competing interests.
