## [Peer Review File · Nature Communications]

Reviewers' comments:

Reviewer #1 (Remarks to the Author); expert in glioblastoma:

Intriguing paper that performs MS classification of a small set of glioma tumors, generates a number of intriguing observations regarding utility of proteomic vs DNaseq vs RNAseq data, in addition to conclusions regarding prognosis, metabolome, cell of origin and response to therapy, offering no future biological validation for any of these conclusions. While the paper is generally sound, it would be strengthened by inclusion of biological experiments or validation set proteomic data that validate claims. At a minimum, they could analyze tumors from a PDX models to see how they fit the classifier, and then use them to validate metabolic and therapeutic claims.

Reviewer #2 (Remarks to the Author); expert in proteomics and bioinformatics in cancer:

Authors

This manuscript from Korea is a very well-written and significant pharmaco-proteogenomic analysis of the highly malignant and common form of glioblastoma distinguished as IDH wild-type. In sum, multi-omics features of 39 IDH wt GBM tumors were compared with 2 IDH mutant GBM and 9 low-grade gliomas. Two clusters were identified: GPC1 with enhanced glycolysis (Warburg effect), neural stem-cell markers, immune checkpoint ligands, and FKBP9, a biomarker for poor prognosis; and GPC2 with high ox-phos proteins, oligodendrocyte and astrocyte biomarkers, and PHGDH, a biomarker for relatively better prognosis. Patient-derived cells (PDCs) were prepared for testing with many chemotherapeutic drugs. An mTORC1/2 dual inhibitor showed notable cytotoxicity against the GPC1 poor prognosis PDCs. This categorization may yield much more useful translation than the traditional four subtypes: classical, mesenchymal, proneural, and neural (line 60) based on transcriptome profiling by TCGA.

The GPC2 protein expression patterns were much more like normal brain than were the GPC1 tumors. With extremely limited data, 2/3 tumors that were recurrent after initially effective therapy switched from GPC1 to GPC2 patterns, a good sign. These GPC subtypes were independent of the four RNA subtypes, though mutation analysis showed EGFRvIII splice isoform and PIK3CA mutations exclusively in GPC1 tumors (of course, there were only 2 mutant GBMs and 9 low-grade gliomas. That should be acknowledged.

I would be interested to know whether other striking findings related to splice isoforms were revealed, such as PKM isoforms (driver of Warburg effect) in Figure 3(d) (line 835).

The single cell analyses yielded interesting results, too.

An important finding was that the proteomic analyses were much more informative than mRNA analyses for capturing action of the drugs. This is quite predictable for kinase-related pathways, which require phosphoproteomic analysis. AZD2014 may be a high-quality lead drug candidate. Equally striking is that only 3 of 271 gene expression biomarkers from RNA studies (presumably meaning up or down regulated compared to normal brain) were "validated" in this proteogenomic study.

The speculations about cross-talk and stem cells are of interest, too.

Methods Details

Line 566: identify the reference protein database used.

Line 573-74: give numbers of peptides and PSMs; why only peptides and PSM controlled with FDR <0.01; the protein FDR should be stated and controlled below 0.01.

Line 576: why do you accept "one-hit wonders" for identification after saying in 575 that the minimum number of uniquely-mapping peptides was set to 2?

Line 612: why did you use and not update the SwissProt version 2014/03? For human studies, it is

generally more convenient to use the sister database neXtProt, updated at least annually.
Line 616: What is the date of the Ensembl 75?
Line 858: typo, "2-hydroxyglutarate".

Gilbert S. Omenn, MD, PhD

Reviewer #3 (Remarks to the Author); expert in glioblastoma:

Manuscript NCOMMS-19-30472:

The paper by Sejin Oh et al. addresses an unmet clinical need, e.g. how to stratify patients with IDH-WT glioblastoma (GBM) for individualized treatment. They approached this using quantitative proteomic analysis on 39 IDH wild-type glioblastoma (GBMs) as well as 2 IDH mutant GBM, 9 low grade glioma and 4 normal brains. They then integrated their findings with their prior published genomic and drug-sensitivity results from the same patients, which they call "integrated pharmaco-proteogenomics". The manuscript is extremely well written overall and mostly based on bioinformatics analyses. These analyses derive potentially interesting protein markers for certain biological functions, but these are not experimentally validated, so the tentative conclusions are forward looking.

The main proposed finding of the paper is the identification of two subgroups of IDH wt GBMs based on proteomic patterns. GBM proteomic cluster 1 (GPC1) tumors exhibit Warburg-like features, and have high expression of neural stem-cell markers and immune checkpoint ligands. GPC1 have a high expression of FKBP prolyl isomerase 9 (FKBP9), which correlates with poor prognosis. GPC2 tumors show high levels of oxidative phosphorylation-related proteins, and differentiated oligodendrocyte and astrocyte markers. GPC2 have high phosphoglycerate dehydrogenase (PHGDH), which correlates with favorable prognosis. To try to connect this new stratification system to subgroup-specific treatments, the authors re-analyzed their prior published results of drug sensitivity testing on primary cultures of the same patients, which evidenced GPC1 cells are most sensitive to mTORC1/2 dual inhibitor AZD2014. They conclude that their analyses can guide GBM prognosis and precision treatment strategies.

While interesting and clearly patient-relevant, the study novelty is somewhat incremental and the conclusions are not yet definitive. The purpose of dividing patients into subtypes is to estimate differences in prognosis or derive subtype-specific treatments. The advantage of this novel proteomics-based subtype classification is not fully demonstrated in their study. Individual markers related to prognosis were proposed from the proteomics analysis, but it is unclear whether this classification in two separate proteomic groups (GPC1 and GPC2) is useful for estimating patient prognosis. The authors show difference in molecular characteristics between the two groups, as well as differences in drug sensitivity on primary tumor cell cultures. However, there is no logical mechanistic connection between the molecular characteristics and drug sensitivity profiles.

IDH wt GBM are known to be divided into multiple subtypes with various genetic abnormalities and RNA expression profiles, so how to reconcile this prior data with only two proteomic subtypes is unclear and perhaps related to the small number of samples analyzed by proteomics (39 GBM), versus TCGA that used 500 GBM samples. Moreover, they showed that IDH mutant GBM and normal brain are included into their GPC2 subtype, indicating the limitation of this subtype. Below are some suggestions the authors can consider to further improve their manuscript.

Major comments:

1) They detected an average of 6,294 proteins and 2,796 phosphorylation sites from their proteomics analysis. In Ext Fig. 1C, they showed that the number of quantified proteins from each set is highly variable. They detected about 5,500 phospho sites in set #8, but only 1,400 in set

#2. Why such difference?

2) In Fig. 1F, they found that IDH wt gliomas have elevated phosphorylation of STAT1 at serine-727 (active STAT1) and its target proteins. Was this elevation found in both GPC subgroups? Confirming the findings in in vitro experiments with pertinent tumor cells to validate the results of proteomics analysis would strengthen the manuscript.

3) In Fig.2b, the heat map shows the expression pattern of GPC1 and 2. Most of GPC1 tumors have very similar pattern, but the pattern of GPC2 tumors is more heterogeneous. The samples in 3rd, 4th and 6th columns from the left of GPC2 differ in pattern from other GPC2 samples and are more similar to GPC1. This data may indicate that additional subtypes may still exist within the GPC2 subtype that may be revealed upon increase in sample numbers.

4) In Fig.2b, the proteomic expression pattern of normal brain is very similar to the pattern of GPC2. In contrast, in RNA seq analysis, there are clear differences between normal and tumors. Why is this?

5) The authors suggest that GPC1 tumors metabolically rely on the Warburg effect based on metabolic enzyme expression levels in tumor tissue. Demonstration of relevant metabolite levels in GPC1 vs. GPC2 tumors and/or primary cells would consolidate the findings.

6) Showing survival curves differ between patients with GPC1 and GPC2 signatures would strengthen the usefulness of this subgrouping for GBM IDH-wt.

7) The authors indicate that PHGDH, RFTN2 and FKBP9 could serve as markers to predict patient outcome. Independent validation by immunohistochemistry of relative expression of these proteins in a blinded set of human samples with different survival would consolidate the findings.

8) Are PHGDH, RFTN2 and FKBP9 functional markers? Addition of functional data with overexpression or downregulation in patient-derived cells would help address this issue.

9) The authors show that GPC1 tumors have high levels of neural stem cell markers and GPC2 have high oligodendrocyte and astrocyte markers. From these findings, it is concluded that GPC1 tumors originate from NSCs, whereas GPC2 tumors originate from differentiated oligodendrocytes and astrocytes. However, it is hard to reach that conclusion only based on proteomic profiles as GPC2 could differentiate from NSCs. These data only show that GPC1 have NSC-like marker expression and GPC2 have astro and oligo-like marker expression.

10) The authors applied their proteomic subtype classification to published single-cell data. Of 3,589 single cells, 357 and 428 cells were classified as sGPC1 and sGPC2, respectively. Why did only about 10% of cells classify as GPC1 and 2? What does that mean for their overall proteomic analysis which was based on whole tumor pieces.

11) prior proteomics analyses of gliomas have been performed. The authors did not mention how their results compare to those of those prior studies, using tumor or other fluids like CSF.

Minor comments:

Line 97: "obtained non-uniquely" is a strange terminology. Are these longitudinal samples in same patients or are they multi-region sampling? A table of all patient samples clarifying this would be helpful. Also explaining the adjacent normal samples (line 129) in this table would further clarify.

Line 132: "comparing IDH wt and mutant gliomas" does this mean a mix of GBM and lower grade gliomas?

Line 147: replace "all the IDH.." with "the two IDH..."

Line 148: replace "and LGGs.." with "and the 9 LGGs.."

Line 152/153: this conclusion appears overstated given the heterogeneity within GPC2. More samples are needed for such conclusions.

Line 174: In figure 3a it would be more intuitive if PC1 could be called PC2, as it associates mostly with GPC2 subgroup. PC2 could be renamed PC1 as it associates with GPC1.

Line 218: replace "considered an origin" with "considered a cell of origin"

Line 219: replace "represents" with "has"

Line 219: what does "non-target phosphoproteins" mean? Sentence a bit unclear.

Line 318/319: this statement is somewhat speculative without experimental validation. Rephrase.

Line 324-326: add reference

Line 338: "primarily driven by" is speculative. Replace with "primarily characterized by"

Line 354: add reference after "for GBM"

Line 359/360: this conclusion is speculative and not supported by experimental data.
Line 361: the use of the word "cross-talk" appears inadequate here.
Line 362: replace "carried" with "carry"

Reviewers' comments:

Reviewer #1 (Remarks to the Author); expert in glioblastoma:

Intriguing paper that performs MS classification of a small set of glioma tumors, generates a number of intriguing observations regarding utility of proteomic vs DNaseq vs RNAseq data, in addition to conclusions regarding prognosis, metabolome, cell of origin and response to therapy, offering no future biological validation for any of these conclusions.

Referee point 1: While the paper is generally sound, it would be strengthened by inclusion of biological experiments or validation set proteomic data that validate claims. At a minimum, they could analyze tumors from a PDX models to see how they fit the classifier, and then use them to validate metabolic and therapeutic claims.

Response: We appreciate this referee's constructive suggestion. In our revised manuscript, we made substantial updates, particularly on experimental validation of our major findings. These include 1) validation of elevated phospho-STAT1 in IDH wild-type GBM using IDH wild-type and mutant cells in an isogenic background (See Referee point 9), 2) multiplex immunohistochemistry data validating the prognostic power of the protein marker PHGDH (See Referee point 14), 3) functional validation of PHGDH using established cell line models assayed in 3D culture conditions with and without relevant genetic and/or chemical perturbations (See Referee point 15), 4) validation of enhanced lactate secretion (Warburg effect) in GPC1 subtype using an independent large scale CCLE metabolomics dataset (See Referee point 12), and 5) validation of intratumoral heterogeneity at a single cell level using independent IDH wild-type GBM tumors via multiplex immunohistochemistry method (See Referee point 17).

Notably, however, we were not able to use PDX models, since, as the referee might expect, amplifying and testing different PDX models in a given revision timeline was not feasible. Therefore, we decided to use alternative validation methods using in vitro experimental models and independent GBM tissues. We hope the referee agrees that the validation results provided in the revision have significantly strengthened our claims.

Reviewer #2 (Remarks to the Author); expert in proteomics and bioinformatics in cancer:

Authors

This manuscript from Korea is a very well-written and significant pharmaco-proteogenomic analysis of the highly malignant and common form of glioblastoma distinguished as IDH wild-type. In sum, multi-omics features of 39 IDH wt GBM tumors were compared with 2 IDH mutant GBM and 9 low-grade gliomas. Two clusters were identified: GPC1 with enhanced glycolysis (Warburg effect), neural stem-cell markers, immune checkpoint ligands, and FKBP9, a biomarker for poor prognosis; and GPC2 with high ox-phos proteins, oligodendrocyte and astrocyte biomarkers, and PHGDH, a biomarker for relatively better prognosis. Patient-derived cells (PDCs) were prepared for testing with many chemotherapeutic drugs. An mTORC1/2 dual inhibitor showed notable cytotoxicity against the GPC1 poor prognosis PDCs. This categorization may yield much more useful translation than the traditional four subtypes: classical, mesenchymal, proneural, and neural (line 60) based on transcriptome profiling by TCGA.

The GPC2 protein expression patterns were much more like normal brain than were the GPC1 tumors. With extremely limited data, 2/3 tumors that were recurrent after initially effective therapy switched from GPC1 to GPC2 patterns, a good sign. These GPC subtypes were independent of the four RNA subtypes, though mutation analysis showed EGFRvIII splice isoform and PIK3CA mutations exclusively in GPC1 tumors (of course, there were only 2 mutant GBMs and 9 low-grade gliomas. That should be acknowledged.

I would be interested to know whether other striking findings related to splice isoforms were revealed, such as PKM isoforms (driver of Warburg effect) in Figure 3(d) (line 835).

The single cell analyses yielded interesting results, too.

An important finding was that the proteomic analyses were much more informative than mRNA analyses for capturing action of the drugs. This is quite predictable for kinase-related pathways, which require phosphoproteomic analysis. AZD2014 may be a high-quality lead

drug candidate. Equally striking is that only 3 of 271 gene expression biomarkers from RNA studies (presumably meaning up or down regulated compared to normal brain) were “validated” in this proteogenomic study.

The speculations about cross-talk and stem cells are of interest, too.

Referee point 2: I would be interested to know whether other striking findings related to splice isoforms were revealed, such as PKM isoforms (driver of Warburg effect) in Figure 3(d) (line 835).

Response: We appreciate this referee’s valuable comments. In our proteome data, a total of 18 protein isoform groups, covering 9 genes (*CAPZB*, *EPB41L3*, *IKBIP*, *MAP2*, *MAP4*, *PFN2*, *PKM*, *RTN1* and *SNX32*) showed different quantification levels between isoforms (Table below).

Gene Symbol	Protein isoform	T-test P value	FDR	Mean GPC1 level	Mean GPC2 level
PKM	P14618;P14618-3 (PKM2)	1.20E-4	3.10E-4	0.17	-0.67
PKM	P14618-2 (PKM1)	3.00E-5	9.70E-5	-0.46	1.10
IKBIP	Q70UQ0-3	1.80E-5	9.70E-5	0.46	-0.60
IKBIP	Q70UQ0-4	3.20E-5	9.70E-5	0.43	-0.47
SNX32	Q86XE0-2	0.036	0.054	0.46	-0.23
SNX32	Q86XE0	0.14	0.17	0.19	-0.54
EPB41L3	Q9Y2J2-3;Q9Y2J2-2	2.10E-6	3.80E-5	-0.34	0.66
EPB41L3	Q9Y2J2	1.70E-4	3.70E-4	-0.27	1.00
RTN1	Q16799-3	2.00E-5	9.70E-5	-0.51	1.20
RTN1	Q16799;Q16799-2	8.20E-4	1.50E-3	-0.30	0.51
CAPZB	P47756	2.20E-3	3.60E-3	-0.15	0.38
CAPZB	P47756-2	0.18	0.20	0.091	-0.073

PFN2	P35080	3.20E-5	9.70E-5	-0.55	0.99
PFN2	P35080-2	0.67	0.71	-0.076	0.020
MAP2	P11137;P11137-2;P11137-3	1.90E-4	3.70E-4	-0.40	0.93
MAP2	P11137-4	0.94	0.94	0.095	0.11
MAP4	P27816-3	0.056	0.078	-0.073	0.24
MAP4	P27816;P27816-2;P27816-6	0.14	0.17	0.011	-0.28

Of the 18 isoform groups, only PKM isoforms exhibited a mutually exclusive expression pattern in two GPC subtypes: PKM1 was significantly elevated in GPC2, while PKM2 was in GPC1. This GPC subtype dependent expression of PKM isoforms supports our hypothesis that GPC1 tumors are more glycolytic and Warburg-like, whereas GPC2 tumors favor oxidative phosphorylation. This is because PKM isoforms function as a key metabolic switch determining the catabolic pathway of pyruvate, either to glycolysis (by PKM2) or to oxidative phosphorylation (by PKM1) (Dong et al. *Oncology Letters* 2016). In our revised manuscript, we updated the Methods to include whole protein isoform analysis and provided a list of protein isoforms found to be GPC subtype selective as follows:

“In total, 18 protein isoforms (sharing a gene symbol) were quantified across the GISs and had different quantification values; these corresponded to nine genes (*CAPZB*, *EPB41L3*, *IKBIP*, *MAP2*, *MAP4*, *PFN2*, *PKM*, *RTN1* and *SNX32*). Among the 18 isoform groups, only PKM isoforms exhibited mutually exclusive expression patterns in two GPC subtypes: PKM1 was significantly elevated in GPC2, whereas PKM2 was in GPC1. For the comparison of PKM isoforms, we selected peptides belong either to PKM1-specific exon 9 or PKM2-specific exon 10. The peptide intensities were normalized based on the corresponding GISs and transformed to the log₂ scale. The average peptide expression values were then calculated according to the specific sequences for either PKM1 or PKM2. Five *PKM1* isoforms (ENST00000319622, ENST00000389093, ENST00000565154, ENST00000565184, and ENST00000568459) and one *PKM2* isoform (ENST00000335181) were used. The average expression of the isoforms was used for PKM1.”

Methods Details

Referee point 3: Line 566: identify the reference protein database used.

Response: As requested by the referee, we clarified the sentences in the Methods section as follows:

“Peptide and protein identification and quantification were performed using MaxQuant⁶⁶ 1.5.6.0. The mass spectrometry raw files were searched against the Swiss-Prot human database (released in March 2014; <http://www.uniprot.org>) using the Andromeda search engine included in MaxQuant.”

Referee point 4: Line 573-74: give numbers of peptides and PSMs; why only peptides and PSM controlled with FDR <0.01; the protein FDR should be stated and controlled below 0.01.

Response: Answered in Referee point 5 below.

Referee point 5: Line 576: why do you accept “one-hit wonders” for identification after saying in 575 that the minimum number of uniquely-mapping peptides was set to 2?

Response: We apologize for the insufficient description and errors in the Methods section. Regarding point 4 above, we did actually apply the same stringency threshold for protein quantification (FDR < 0.01). Also, we did analyze proteins whose existence are supported by at least two unique peptides (≥ 8 amino acids in length with no miscleavages allowed). We comprehensively revised the paragraph in the Methods section as follows:

“The following MaxQuant search parameters were used: semispecific trypsin was selected as the enzyme; the carbamidomethylation of cysteine was set as a fixed modification; N-terminal protein acetylation and oxidation (M) were set as variable modifications; and phosphorylation (STY) was set as a variable modification for phosphorylation-enriched samples. The reporter ion was set as six-plex TMT for quantification. Peptide matches were filtered by a minimum length of eight amino acids and no miscleavages were allowed. The false discovery rate (FDR) was set to 0.01 at both the protein and peptide spectrum match

(PSM) levels. Proteins identified by at least two unique peptides were used. For protein quantification, the minimum ratio count was set to two, and the peptide for protein quantification was set as unique. Other settings were kept at their default values. In total, 9,367 protein groups, 179,234 stripped peptides, and 2,750,407 peptide spectral matches (PSMs) were identified from the global proteome. In the case of the phosphoproteome, 8,019 phosphorylation sites, 16,377 phosphorylated peptides, and 276,153 PSMs were identified. The mass spectrometry proteomics data have been deposited to the ProteomeXchange Consortium (<http://proteomecentral.proteomexchange.org>) via the PRIDE partner repository with the dataset identifier PXD015545.”

Referee point 6: Line 612: why did you use and not update the SwissProt version 2014/03? For human studies, it is generally more convenient to use the sister database neXtProt, updated at least annually.

Response: In our integrated analysis, as we had to generate personalized databases that reflect sample-specific somatic mutations, we had to freeze the reference genome and transcriptome and proteome databases at the time point of when we generated mass spectrometry data. This is why those databases were somewhat outdated. However, the differences in the identified peptides and proteins between the old and new releases are small. To evaluate the differences therein, we used the raw data of set1 samples (4-1, 12, 18, 20, and 655N) and compared the qualitative and quantitative results of the two databases (March 2014 and Sept 2019). Qualitatively, 99% of the discovered peptides (Figure a below) and proteins (Figure b below) were identical.

Only 1% of the proteins and peptides unique to either database did not affect any of the main findings of our study. Furthermore, the different quantitative values generated from the two databases, observed only in 166 out of 6465 protein groups, exhibited almost perfect correlation (Figure below), confirming again that relative protein expression levels are not affected by the database issue.

However, we appreciate the referee's suggestion and agree that neXtProt is now a better option for proteomic studies. Although we did not use it for this manuscript, we will definitely consider using it in the future.

Referee point 7: Line 616: What is the date of the Ensembl 75?

Response: It was released in February 2014. We added this information in the Methods section as follows:

“For this database, the transcript models in Ensembl 75 (released in February 2014) whose FPKM values were > 1 were used, similar to our previous study.”

Referee point 8: Line 858: typo, “2-hydroxyglutarate”.

Response: Corrected. Thanks.

Gilbert S. Omenn, MD, PhD

Reviewer #3 (Remarks to the Author); expert in glioblastoma:

Manuscript NCOMMS-19-30472:

The paper by Sejin Oh et al. addresses an unmet clinical need, e.g. how to stratify patients with IDH-WT glioblastoma (GBM) for individualized treatment. They approached this using quantitative proteomic analysis on 39 IDH wild-type glioblastoma (GBMs) as well as 2 IDH mutant GBM, 9 low grade glioma and 4 normal brains. They then integrated their findings with their prior published genomic and drug-sensitivity results from the same patients, which they call “integrated pharmaco-proteogenomics”. The manuscript is extremely well written overall and mostly based on bioinformatics analyses. These analyses derive potentially interesting protein markers for certain biological functions, but these are not experimentally validated, so the tentative conclusions are forward looking.

The main proposed finding of the paper is the identification of two subgroups of IDH wt GBMs based on proteomic patterns. GBM proteomic cluster 1 (GPC1) tumors exhibit Warburg-like features, and have high expression of neural stem-cell markers and immune

checkpoint ligands. GPC1 have a high expression of FKBP prolyl isomerase 9 (FKBP9), which correlates with poor prognosis. GPC2 tumors show high levels of oxidative phosphorylation-related proteins, and differentiated oligodendrocyte and astrocyte markers. GPC2 have high phosphoglycerate dehydrogenase (PHGDH), which correlates with favorable prognosis. To try to connect this new stratification system to subgroup-specific treatments, the authors re-analyzed their prior published results of drug sensitivity testing on primary cultures of the same patients, which evidenced GPC1 cells are most sensitive to mTORC1/2 dual inhibitor AZD2014. They conclude that their analyses can guide GBM prognosis and precision treatment strategies.

While interesting and clearly patient-relevant, the study novelty is somewhat incremental and the conclusions are not yet definitive. The purpose of dividing patients into subtypes is to estimate differences in prognosis or derive subtype-specific treatments. The advantage of this novel proteomics-based subtype classification is not fully demonstrated in their study. Individual markers related to prognosis were proposed from the proteomics analysis, but it is unclear whether this classification in two separate proteomic groups (GPC1 and GPC2) is useful for estimating patient prognosis. The authors show difference in molecular characteristics between the two groups, as well as differences in drug sensitivity on primary tumor cell cultures. However, there is no logical mechanistic connection between the molecular characteristics and drug sensitivity profiles.

IDH wt GBM are known to be divided into multiple subtypes with various genetic abnormalities and RNA expression profiles, so how to reconcile this prior data with only two proteomic subtypes is unclear and perhaps related to the small number of samples analyzed by proteomics (39 GBM), versus TCGA that used 500 GBM samples. Moreover, they showed that IDH mutant GBM and normal brain are included into their GPC2 subtype, indicating the limitation of this subtype. Below are some suggestions the authors can consider to further improve their manuscript.

Major comments:

Referee point 8: They detected an average of 6,294 proteins and 2,796 phosphorylation

sites from their proteomics analysis. In Ext Fig. 1C, they showed that the number of quantified proteins from each set is highly variable. They detected about 5,500 phospho sites in set #8, but only 1,400 in set #2. Why such difference?

Response: We appreciate this referee's valuable comments. The variation the referee pointed out is caused by different sample preparation methods used for samples in the first batch (consisting of set 1 to set 6) and the second batch (consisting of set 7 to set 11). The Methods section described this as follows:

"The combined sample comprising the first batch was dried *in vacuo* and subsequently desalted using HLB-SPE. By contrast, the combined sample for the second batch was directly desalted and subsequently dried *in vacuo* because drying samples in the presence of hydroxylamine is detrimental to phospho-peptides."

We changed the sample preparation method for the second batch to improve the stability of phospho-peptides. As shown in the Extended Data Fig. 1C, the change allowed us to identify twice as many phospho-peptides in the second batch. This change only affected the sensitivity of detection and did not affect phospho-protein levels, since the levels were always normalized by GIS controls. We demonstrated this by showing that samples from different batches were evenly mixed in the principal component analysis (PCA) with 548 phospho-sites quantified across all 11 sets in the two batches (Figure below). If there was a strong batch effect, samples on the PCA plot might be clustered by their own batches.

Referee point 9: In Fig. 1F, they found that IDH wt gliomas have elevated phosphorylation of STAT1 at serine-727 (active STAT1) and its target proteins. Was this elevation found in both GPC subgroups? Confirming the findings in in vitro experiments with pertinent tumor cells to validate the results of proteomics analysis would strengthen the manuscript.

Response: We appreciate this feedback. As the referee suggested, we confirmed our findings of IDH wild-type selective elevation of STAT1 (pSer727) with an isogenic pair of pertinent GBM cells obtained from ATCC: IDH1 wild-type (U87MG) and IDH1 knock-in derivative (R132H). Coherent to our original findings, STAT1 (pS727) levels were 1.5 fold higher in *IDH1* wild-type cells than in its mutant derivative (Figure below; fold difference was estimated using ImageJ) (Figure below). We added this result to the revised manuscript (new Extended Data Fig. 1f) as follows:

“Also, *IDH* mutation status directly affected STAT1-pS727 levels, as shown in an *IDH* wild-type and mutant GBM cell line pair in an isogenic U87MG background (Extended Data Fig. 1f).”

With regard to the referee's question, there was no meaningful difference in STAT1-pS727 levels between the two subtypes in our SMC1 cohort (Figure below), indicating it is associated with IDH mutant status not GPC subtype.

Referee point 10: In Fig.2b, the heat map shows the expression pattern of GPC1 and 2. Most of GPC1 tumors have very similar pattern, but the pattern of GPC2 tumors is more heterogeneous. The samples in 3rd, 4th and 6th columns from the left of GPC2 differ in pattern from other GPC2 samples and are more similar to GPC1. This data may indicate that additional subtypes may still exist within the GPC2 subtype that may be revealed upon increase in sample numbers.

Response: As the reviewer pointed out, the expression pattern of GPC2 was relatively more heterogeneous than that of GPC1. However, we would like to emphasize that two of the three samples the referee pointed out (3-1 and 3-2) are more closely associated with GPC2 than GPC1, as demonstrated by significantly high consensus index values for GPC2 (Figure below; classification frequency estimated from 1,000X permutation resampling, **; Wilcoxon rank-sum, $P < 0.00001$). However, 5-1 is an outlier that was associated with GPC2-subtype samples at a slightly higher frequency (52% of GPC2 samples vs. 51% of GPC1 samples) (Figure below).

This displays the ambiguous expression pattern of proteins associated with two GPC subtypes (Figure below).

This suggests that, as the referee pointed out, we cannot rule out the possibility that increased sample size may lead to an additional subtype distinct from the two GPC subtypes. However, based on our analysis of 39 samples of IDH wild-type GBM, we are confident that GPC1 and GPC2 may still represent major IDH wild-type GBM subtypes, even with increased sample size.

To address the referee’s important point, we added the following sentence in the Discussion section of the revised manuscript:

“However, some of the samples, particularly in GPC2, exhibited heterogeneous expression patterns, compared to other samples in the subtype, suggesting that increased sample size may lead to additional subtype(s) distinct from the two major GPC subtypes.”

Referee point 11: In Fig.2b, the proteomic expression pattern of normal brain is very similar to the pattern of GPC2. In contrast, in RNA seq analysis, there are clear differences between normal and tumors. Why is this?

Response: We apologize for the unclear labels on the Figure. Normal tissues on the left were only used for proteomic analysis not for RNA-seq. Therefore, we never compared normal and tumor tissues at the RNA level. In the figure, grey stripes indicated normal controls, rather than molecular subtypes. In the revised manuscript, we clarified the figure as follows:

Referee point 12: The authors suggest that GPC1 tumors metabolically rely on the Warburg effect based on metabolic enzyme expression levels in tumor tissue. Demonstration of relevant metabolite levels in GPC1 vs. GPC2 tumors and/or primary cells would consolidate the findings.

Response: As the referee suggested, showing the actual metabolite levels in primary tumors and/or primary cells would be ideal in the validation of the functional consequence of our proteome data. However, we hope the referee understands that this would require both a large amount and a large number of samples to get reliable measurement of metabolites and to obtain sufficient statistical power for comparison, respectively. As available fresh-frozen tissues were limited and patient derived cells grow very slowly, we decided to look into recent large scale metabolomics datasets instead in an attempt to satisfy both requirements. Therein, 225 metabolites (including lactate) were profiled for 928 pertinent cell lines of different lineages, including GBM (Li et al. Nat Med 2019). We used transcriptome data to

determine sGPC subtypes of 47 GBM cell lines (Figure a below) and compared their lactate levels (Figure b below). Coherent to our findings, sGPC1 GBM cell lines (classified under permutation resampling $p < 0.05$, $N = 6$) exhibited higher levels of lactate than sGPC2 lines ($N = 10$). This supports our hypothesis that GPC1 tumors may rely on the Warburg effect, with GPC2 relying on oxidative phosphorylation. Although it is beyond the scope of this study, as the referee suggested, additional large scale studies of primary GBM tumors with intact tumor microenvironment are needed for ultimate validation.

In the revised manuscript, we added the findings from our revision experiments to the Results section as follows:

“Coherent with our proteomic data, GBM cell lines belonging to gene expression-based surrogate-GPC1 subtype (sGPC1) exhibited higher lactate levels (Fig. 3e) in the analysis of cancer cell line encyclopedia metabolomics data.”

Referee 13: Showing survival curves differ between patients with GPC1 and GPC2 signatures would strengthen the usefulness of this subgrouping for GBM IDH-wt.

Response: When we compared the survival of the GPC1 and GPC2 samples in the Samsung Medical Center (SMC) 1 cohort (used for proteomic analysis), GPC1 samples showed slightly worse prognosis, although the difference was not statistically significant (Figure a below). This trend was observed as well in other independent GBM cohorts with long term survival follow-up, including our Yonsei cohort (Figure b below, Park et al. Sci Rep 2019)

and the Association des Neuro-Oncologue d'Expression (ANOCEF, Ducray et al. Mol Cancer 2010) cohort (Figure c below) (Figure below). However, log rank test P values were all insignificant, as indicated below.

Although GPC subtype itself does not directly predict patient survival, we like to emphasize that some of the protein markers that are significantly associated with the subtypes have robust predictive power of prognosis for *IDH* wild-type GBM (See Figure 4a & 4b, Response 14 below). To make this clear, we added the following sentence to the Results section:

“However, GPC subtypes did not directly show a significant difference in prognosis (Log rank test $P = 0.0548$, data not shown).”

Response 14: The authors indicate that PHGDN, RFTN2 and FKBP9 could serve as markers to predict patient outcome. Independent validation by immunohistochemistry of relative expression of these proteins in a blinded set of human samples with different survival would consolidate the findings.

Response: We are grateful for the reviewer’s constructive suggestion. To address the request, we approached this in two ways: first was using gene expression based classification, followed by survival analysis of other independent GBM datasets, and second was using in-house tumor microarray (SMC-TMA) analysis to measure relative expression of these proteins by immunohistochemistry, as the referee suggested.

The first approach was feasible since expression of all three proteins exhibited strong positive correlations with their mRNA levels in our SMC1 cohort samples (Figure below).

Using the same method by which these survival markers were originally identified in the TCGA dataset (Uhlen et al. Science 2017; i.e. divide patients at the gene expression cutoff providing the most significant difference in prognosis), we classified patients in the Yonsei and ANOCEF cohorts into PHGDH high vs. low, RFTN2 high vs. low, and FKBP9 high vs. low, respectively, based on gene expression values of the three markers. As shown in the Figure below, *PHGDH* and *RFTN2* showed consistent results to our observation in both cohorts (i.e. elevated expression is associated with good prognosis). However, *FKBP9*, a poor prognostic marker confirmed in our study at the protein level, was reproduced in ANOCEF, but not in the Yonsei cohort.

The differences in prognosis did not rely on single bifurcation cutoff value of gene expression, as shown in the scanning window plot below (blue: good prognosis in high expression group, orange: poor prognosis in high expression group). Meanwhile, a prognostic difference was consistently observed across broad range of cutoffs, except for FKBP9, in the Yonsei dataset.

Second, we used a tumor microarray (SMC-TMA) to measure relative expression of these proteins by Opal, a method for multiplex fluorescent immunohistochemistry. The SMC-TMA contained an array of tumor cores taken from paraffin-embedded tumor tissues and consisted of 128 tissue samples: 83 IDH-wild-type GBMs, 4 IDH-mutant GBMs, 35 low grade gliomas, and 6 normal control normal tissues. Also, the array included 8 SMC1 and 15 SMC2 samples. Due to limited availability of SMC-TMA slides and limited slots for antibodies for multiplexing, PHGDH was selected over RFTN2 as a favorable prognostic marker based on its greater hazard ratios and more significant Cox regression P values. FKBP9 was also not included because it showed inconsistent prognosis patterns in the independent datasets, as shown above, and because commercial antibodies available for IHC studies were unable to generate reliable signals in the pre-optimization step. Coherently, in both TCGA and SMC1, PHGDH-elevated IDH wild-type GBM patients showed better prognosis than PHGDH-low patients (Figure below).

In the revised manuscript, we added these findings to the Result section as follows:

“Of the three proteins, PHGDH was the strongest biomarker (Univariate Cox $P = 0.0071$) associated with long-term survivors of *IDH* wild-type GBM patients in the SMC1 cohort (Fig. 4d), as well as in other independent datasets at the mRNA-level (Extended Data Fig. 4a). Favorable prognosis of PHGDH-high tumors was further validated in 42 independent *IDH* wild-type GBM tumors assessed by immunohistochemistry on a tumor tissue microarray (SMC-TMA) using an anti-PHGDH antibody (Fig.4g).” We also updated Method section to include the SMC-TMA study.

Referee point 15: Are PHGDH, RFTN2 and FKBP9 functional markers? Addition of functional data with overexpression or downregulation in patient-derived cells would help address this issue.

Response: We appreciate this feedback, and it has been one of the major focuses of our post-submission efforts. To address the referee’s comments, we prepared an experimental setup to test whether genetic and/or chemical perturbation of PHGDH in relevant patient-derived cells affects aggressiveness. Here, in our revision study, we focused on PHGDH since only this marker was consistently associated with prognosis in multiple independent cohorts (Referee point 14). We used GBM patient-derived established cell line models rather than patient-derived primary cells (PDCs) since PDCs are not ideal experimental models for genetic perturbation due to their slow proliferation, suspension growth, and limits in number of cell divisions. Our strategy was to find relevant cell line models based on protein marker

expression in steady state growth conditions and to subject them into Matrigel-based 3D sphere culture conditions for quantitative evaluation of the phenotype.

First, we could divide nine tested GBM cell lines into Nestin-high (potentially representing GPC1 subtype) and PHGDH-high (potentially representing GPC2 subtype) groups (Figure a below). PHGDH enzymatic activity was concordantly higher in the four PHGDH-high cell lines (Figure b below). Notably, FKBP9 expression levels were correlated with Nestin (Figure a below), consistent with our findings in primary tumors.

Next, we selected two PHGDH-high cell lines, HS683 and SNU1105, and treated them with NCT-502, a selective PHGDH inhibitor. We observed significantly increased invasion (invasive front of the protrusions are indicated by yellow arrowheads in Figure c below) from both spheres in 3D matrix (Figure b below) at a concentration suppressing enzymatic activity (Figure a below).

HS683

SNU1105

Conversely, overexpression of PHGDH in two PHGDH negative cell lines, SNU201 and A172, decreased invasion (Figure below).

Taken together, these data indicate that PHGDH is a functional marker and suggest that PHGDH may prolong patient survival by suppressing tumor invasion via its increased enzymatic activity. The revised manuscript now includes new Fig 4g and 4h and new Extended Data Fig 4a for the analysis of the results of functional studies of PHGDH. In accordance therewith, substantial updates were made to the Results as follows:

“The good prognosis of the PHGDH-high group suggests a functional role for PHGDH in limiting tumor aggressiveness. Intriguingly, NCT-502, a chemical inhibitor of PHGDH, significantly increased invasion of tumor spheres (Fig. 4h) derived from PHGDH-active GBM cell lines (Extended Data Fig. 4b & 4c, Supplementary Data 5) into 3D matrix. Conversely, PHGDH overexpression in PHGDH-deficient GBM cell lines decreased invasion (Extended Data Fig. 4d), suggesting that PHGDH may prolong patient survival by suppressing tumor invasion via its increased enzymatic activity.” We also updated the Methods section to include the 3D sphere invasion assay and made some minor modifications throughout the manuscript to reflect the findings.

Referee point 16: The authors show that GPC1 tumors have high levels of neural stem cell markers and GPC2 have high oligodendrocyte and astrocyte markers. From these findings, it is concluded that GPC1 tumors originate from NSCs, whereas GPC2 tumors originate from differentiated oligodendrocytes and astrocytes. However, it is hard to reach that conclusion only based on proteomic profiles as GPC2 could differentiate from NSCs. These data only show that GPC1 have NSC-like marker expression and GPC2 have astro and oligo-like marker expression.

Response: We agree with the referee's point that our data are not sufficient to draw a conclusion on cell of origin of the two subtypes. A difference in cell of origin may potentially explain the observation. Alternatively, as the referee speculated, two subtypes may represent cell of origin status and differentiated status therefrom, respectively. With regard to this alternative hypothesis, we showed in Extended Data Fig. 2e that relapsed tumors in the three longitudinal cases tended to be GPC2. Although the sample size was too small to draw a definitive conclusion, these data better fit the alternative hypothesis rather than the different cell-of-origin hypothesis described in the original manuscript. Therefore, we revised the related sentences in the Results and Discussion sections accordingly to avoid any misinterpretation or overstatement as follows:

Result: "Together, with the observation that recurrent tumors tend to be GPC2 (Extended Data Fig. 2e), one potential explanation of these data is that GPC1 tumors originate from NSCs, whereas GPC2 tumors differentiate from GPC1. However, we cannot exclude the possibility that GPC2 tumors may originate directly from oligodendrocytes and astrocytes."

Discussion: "A recent study showed that 56% of human GBM cases originate from SVZ-derived GSCs²⁵. Thus, we hypothesized that GPC1 subtype might originate from SVZ-derived glioma stem cells (GSCs)."

We also modified the title of the Results section as follows: "GPC-subtype dependent expression of neural stem cell, oligodendrocyte, and astrocyte markers."

Referee point 17: The authors applied their proteomic subtype classification to published single-cell data. Of 3,589 single cells, 357 and 428 cells were classified as sGPC1 and sGPC2, respectively. Why did only about 10% of cells classify as GPC1 and

2? What does that mean for their overall proteomic analysis which was based on whole tumor pieces.

Response: As the referee pointed out, only a small fraction (22%) of single cells were classified into sGPC subtypes. This is because we selected single cells with a high stringent statistical cut-off (permutation $P < 0.05$) to remove falsely classified cells due to limited mRNA-protein correlation and a sparsity of single cell RNA-seq (scRNA-seq) data. Sparsity (missing values) is one of the most challenging issues in scRNA-seq analysis, making it difficult to assign reliable sGPC subtypes to cells. Indeed, the sGPC-classified cells had statistically higher proportions of expressed genes than the unclassified cells (Wilcoxon rank-sum test $P = 2.32E-7$). With the classified single cells, we showed that individual tumors contain neoplastic single cells of different sGPC subtypes and that a dominant cellular population might determine the representative subtype of bulk tumors.

To further validate the intratumoral heterogeneity of GPC subtypes at a single cell level, we used a tumor microarray (SMC-TMA) of independent *IDH* wild-type GBM tissues and measured relative expression of PHGDH (good prognostic and representing GPC2, Fig. 4c) and Nestin (representing GPC1, Fig. 4e) by multiplex fluorescent immunohistochemistry (Opal), both of which generated reliable signal intensities at a single cell resolution. As shown in the Figure below, consistent to our original findings, sGPC1 tumors (labeled in blue) comprised a significantly higher fraction of Nestin-positive neoplastic cells, whereas sGPC2 tumors (labeled in red) showed a significantly higher fraction of PHGDH-positive neoplastic cells.

Intratumoral heterogeneity, observed from the scRNA-seq analysis, was clearly seen in the TMA results. If you look at the representative images below, both Nestin⁺/PHGDH⁻ cells (colored in red, representing GPC1 subtype) and PHGDH⁺/Nestin⁻ cells (colored in yellow, representing GPC2 subtype) were found in all the tumor cores, although at different ratios matching to their sGPC subtypes (i.e., two sGPC1 tumors contained a higher frequency of Nestin⁺ cells, while two sGPC2 tumors contained a higher frequency of PHGDH⁺ cells).

Collectively, scRNA-seq analysis and multiplex immunohistochemistry analysis of independent TMA datasets clearly demonstrated intratumoral heterogeneity in IDH wild-type GBM tumors and helped highlight the nature of GPC subtypes at a single cell resolution. Our revised manuscript now includes a new Fig. 5g demonstrating single cell level heterogeneity of Nestin⁺ and PHGDH⁺ cells in tumor tissues, and we updated the Results section as indicated below.

“To further validate the intratumoral heterogeneity of GPC subtypes at a single cell level, we used a tumor microarray (SMC-TMA) of independent *IDH* wild-type GBM tissues to measure relative expression of PHGDH (good prognostic marker representing GPC2, Fig 4c) and Nestin (representing GPC1, Fig 4e) by multiplex florescent immunohistochemistry that generated reliable signal intensities at a single cell resolution. Consistent with our findings in the proteomic analysis, sGPC1 tumors exhibited a significantly higher fraction of Nestin positive neoplastic cells, whereas sGPC2 tumors comprised a significantly higher fraction of PHGDH-positive neoplastic cells (Extended Data Fig. 5b). Intratumoral heterogeneity, observed from the scRNA-seq data, was clearly seen in the multiplex florescent immunohistochemistry results. Both Nestin⁺/PHGDH⁻ cells (representing GPC1 subtype) and PHGDH⁺/Nestin⁻ cells (representing GPC2 subtype) were found in all tumor cores, albeit with different ratios matching their sGPC subtypes (i.e., two sGPC1 tumors contained a higher frequency of Nestin⁺ cells, while two sGPC2 tumors contained higher frequency of PHGDH⁺ cells (Fig. 5g).”

Referee point 18: prior proteomics analyses of gliomas have been performed. The authors did not mention how their results compare to those of those prior studies, using tumor or other fluids like CSF.

Response: As the referee pointed out, there are several previous studies reporting proteomic analysis on fluids or glioma tissues. For example, Gahoi et al. analyzed cerebrospinal fluid from high and low grade glioma samples, focusing on differentially secreted proteins from gliomas at different levels (high vs. low grade, IDH wild-type vs. mutant) (Gahoi et al. *Proteomics. Clinical Applications* 2017). More recently, for the early diagnosis and for following the progression of GBM patients, Osti et al. investigated protein compositions in extracellular vesicles in blood (Osti et al. *Clinical Cancer Research* 2019). These studies commonly investigated a small number of diagnostic biomarker proteins with minimally invasive blood sampling, rather than comprehensively profiling global proteomic contents in glioblastomas.

So far, only a few studies have attempted proteomic analysis of GBM tissues. For example, in an effort by the TCGA consortium, Brennan et al. conducted reverse phase protein array (RPPA) by which 214 sample lysates were probed with 171 antibodies targeting major signaling pathway proteins (Brennan et al. *Cell* 2013). More recently, Hutter et al. applied the same method to 60 GBM samples to classify them based on selectively activated signaling pathways (Hutter et al. *J Neu Onc* 2016). However, these RPPA based analyses were strictly limited to the tested antibodies, which are not readily applicable to global proteomic analysis. More recently, a study of mass spectrometry-based global proteomic analysis of eight GBM tissues and paired normal tissues was conducted to identify differentially expressed proteins and pathways in tumor tissue, compared to normal tissue (Song et al. *Oncotarget* 2017). Although they found Nestin and OXPHOS proteins to be dysregulated in GBMs, compared to normal tissue, they were unable to investigate inter- and intra-tumoral heterogeneity, prognostic protein markers, and protein-based drug response predictions, which we believe our manuscript first addresses in GBM. Per the referee's suggestion, we added the following revision to the Introduction section:

“However, although several studies have conducted proteomic analysis of glioma tissue samples or secreted proteins in blood (Gahoi et al. Proteomics. Clinical Applications 2017, Osti et al. Clinical Cancer Research 2019, Song et al. Oncotarget 2017), large-scale proteomic characterization in the context of GBM has not yet been conducted.”

Referee point 19: The authors show difference in molecular characteristics between the two groups, as well as differences in drug sensitivity on primary tumor cell cultures. However, there is no logical mechanistic connection between the molecular characteristics and drug sensitivity profiles.

Response: We greatly appreciate the referee’s attention to the gap in the logical mechanistic connection for Fig. 6c in the original manuscript. Of the six drugs (Tandutinib, Olaparib, Crizotinib, and AZD2014 for GPC1 subtype and Erismodegib and Canertinib for GPC2 subtype), we only provided logical mechanistic connections for Olaparib in Fig. 6d by showing that GPC1-subtype tumors have elevated BRCAness scores and PARP inhibitor sensitivity scores, indicating the concordant pathway activation at the protein level. Upon the referee’s request, we thoroughly investigated the difference in drug target pathway activation status in the two GPC subtypes at the protein level. As shown in the Figure below, we observed drug-sensitivity and target-pathway activation relationships for all other drugs: tandutinib (PDGFR inhibitor), PDGFR_Binding; crizotinib (ALK, MET, ROS1 inhibitor), Oncogenesis_by_MET; olaparib (PARP inhibitor), BRCAness score; AZD2014 (mTORC1/2 dual inhibitor), Translational_Initiation; erismodegib (Hedgehog inhibitor), Hedgehog_GLI_Pathway; and canertinib (pan-ERBB inhibitor), ERBB2_ERBB3_Pathway.

We would like to emphasize that our manuscript already described logical mechanistic connections between the molecular characteristics and drug sensitivity profiles of other drug sensitivities independent of GPC subtypes (i.e., Fig. 6b for panobinostat and bortezomib and Extended Figure 5c/d/e for afatinib, bosutinib, lapatinib, and AZD4547). In result, the revised manuscript now comprises updated Fig. 6d, as well as additions to the Results that incorporate the findings in responding to the reviewer's comments as follows:

“Coherent drug-sensitivity and target-pathway activation relationships for all of these drugs were observed at the protein-levels (Fig. 6d): tandutinib (PDGFR inhibitor), PDGFR_Binding; crizotinib (ALK, MET, ROS1 inhibitor), Oncogenesis_by_MET; olaparib (PARP inhibitor), BRCAness score; AZD2014 (mTORC1/2 dual inhibitor), Translational_Initiation; erismodegib (Hedgehog inhibitor), Hedgehog_GLI_Pathway; and canertinib (pan-ERBB inhibitor), ERBB2_ERBB3_Pathway. Taken together, these data suggest that tandutinib, olaparib, crizotinib and AZD2014 might be a promising targeted therapy for GPC1 tumors and that erismodegib and canertinib might be more promising for GPC2 tumors.”

Minor comments:

Referee point 20: Line 97: “obtained non-uniquely” is a strange terminology. Are these longitudinal samples in same patients or are they multi-region sampling? A table of all patient samples clarifying this would be helpful. Also explaining the adjacent normal samples (line 129) in this table would further clarify.

Response: A table of all patient samples with annotations, including multi-region, longitudinal, and adjacent normal information, was provided in Supplementary Data 1 in the original manuscript. Upon the referee's request, we corrected the unclear sentence as follows:

“20 out of 50 samples were obtained redundantly from multiple regions or at different time points and had different properties regarding mutation, RNA subtype, 5-aminolevulinic acid (5-ALA) positivity, location (locally adjacent or core and margin of tumors), or primary/relapse status (Supplementary Data 1).”

Referee point 21: Line 132: “comparing IDH wt and mutant gliomas” does this mean a mix of GBM and lower grade gliomas?

Response: Yes. In the original Results section, we summarized overall proteomic profiles obtained from 50 glioma samples and 4 normal controls. Therefore, the comparison you pointed out was conducted between 44 *IDH*-wild-type gliomas (39 grade IV and 5 low grade) and 6 *IDH*-mutant gliomas (2 grade IV and 4 low grade). To make this clear, we changed the section title and clarified the sentence as follows:

“Proteomic data represent glioma disease state and underlying biology”, “We then compared *IDH* mutant (N = 6; 2 grade IV and 4 low grade) and *IDH* wild-type (N=44; 39 grade IV and 5 low grade) gliomas.”

Referee point 22: Line 147: replace “all the IDH..” with “the two IDH...”

Response: Corrected as the referee suggested.

Referee point 23: Line 148: replace “and LGGs..” with “and the 9 LGGs..”

Response: Corrected as the referee suggested.

Referee point 24: Line 152/153: this conclusion appears overstated given the heterogeneity within GPC2. More samples are needed for such conclusions.

Response: We revised the sentence and added the following sentence in Discussion:

“However, some of the samples, particularly in GPC2, exhibited heterogeneous expression patterns, compared to other samples in the subtype, suggesting that increased sample size may lead to additional subtype(s) distinct from the two major GPC subtypes.”

Referee point 25: Line 174: In figure 3a it would be more intuitive if PC1 could be called PC2, as it associates mostly with GPC2 subgroup. PC2 could be renamed PC1 as it associates with GPC1.

Response: In our manuscript, PC1 and PC2 refer to the first principal component and the second PC calculated from a statistical method called “principal component analysis (PCA)”. PC1 accounts for the largest variability in the data, and PC2 accounts for as much of the remaining variability as possible, etc. In our results, the largest variability present in our proteome dataset (PC1) was exactly the one that differentiated the two GPC subtypes. Therefore, we hope the referee understands that it is not something that we can rename.

Referee point 26: Line 218: replace “considered an origin” with “considered a cell of origin”

Response: Corrected as the reviewer suggested.

Referee point 27: Line 219: replace “represents” with “has”

Response: Corrected as the reviewer suggested.

Referee point 28: Line 219: what does “non-target phosphoproteins” mean? Sentence a bit unclear.

Response: Changed to “unrelated phosphoproteins.”

Referee point 29: Line 318/319: this statement is somewhat speculative without experimental validation. Rephrase.

Response: Per the referee’s suggestion, we toned down the sentence as follows:

“suggests a possibility that EGFR recycling activity, rather than EGFR kinase activity, may determine responses to lapatinib.”

Referee point 30: Line 324-326: add reference

Response: Done.

Referee point 31: Line 338: “primarily driven by” is speculative. Replace with “primarily characterized by”

Response: Corrected as the reviewer suggested.

Referee point 32: Line 354: add reference after “for GBM”

Response: Done.

Referee point 33: Line 359/360: this conclusion is speculative and not supported by experimental data.

Response: We agree with the reviewer’s comment. In the revised manuscript, we updated the sentence as follows:

“Thus, further research may be needed to evaluate whether the canonical PHGDH function or its promiscuous function is associated with favorable prognosis in *IDH* wild-type GBM.”

Referee point 34: Line 361: the use of the word “cross-talk” appears inadequate here.

Response: Updated to “mechanistic connections.”

Referee point 35: Line 362: replace “carried” with “carry”

Response: Corrected. Thanks.

Reviewer #1 (Remarks to the Author):

Revised manuscript makes some effort to validate findings.

Reviewer #2 (Remarks to the Author):

Authors

You have utilized the review comments and soecific suggestions for clarification and for experimental validation to notably strengthen this manuscript. You report numerous new experiments and analyses from available data resources. Many findings and conclusions have been inserted into the text, and fresh figures and Extended Data materials have been added. It is important that you share with the readers what you have presented to the reviewers.

Two specific details:

Point 12: Surprised that the largest subtype in the figure is "unknown".

Point 18: An important new source of GBM proteogenomics findings should be mentioned from the U.S. National Cancer Institute CPTAC3 project:

https://pdc.cancer.gov/pdc/browse/filters/disease_type:Glioblastoma

December 2019: A Glioblastoma (GBM) discovery cohort of 99 tumor samples and 10 GTEx normal samples analyzed by global proteomic and phosphoproteomic mass spectrometry.

Gilbert S. Omenn, MD, PhD

Reviewer #3 (Remarks to the Author):

We congratulate the authors for adequately responding to our suggestions with additional experiments and manuscript modifications.

REVIEWERS' COMMENTS:

Reviewer #1 (Remarks to the Author):

Revised manuscript makes some effort to validate findings.

Response: We appreciate this referee's valuable comments and suggestions.

Reviewer #2 (Remarks to the Author):

Authors

You have utilized the review comments and specific suggestions for clarification and for experimental validation to notably strengthen this manuscript. You report numerous new experiments and analyses from available data resources. Many findings and conclusions have been inserted into the text, and fresh figures and Extended Data materials have been added. It is important that you share with the readers what you have presented to the reviewers.

Two specific details:

Referee point 12: Surprised that the largest subtype in the figure is “unknown”.

Response: First, we greatly appreciate this referee's insights and valuable suggestions throughout the revision process. The samples were labeled as “unclassified” or “unknown” in our surrogate GPC subtyping because they did not reach our stringent statistical threshold. In the revised manuscript, we removed “unclassified” and “unknown” samples from the Supplementary Fig 3d and Supplementary Fig 5a, respectively, to enhance the clarity of the Figures.

Referee Point 18: An important new source of GBM proteogenomics findings should be mentioned from the U.S. National Cancer Institute CPTAC3 project: https://pdc.cancer.gov/pdc/browse/filters/disease_type:Glioblastoma December 2019: A Glioblastoma (GBM) discovery cohort of 99 tumor samples and 10 GTEx normal samples analyzed by global proteomic and phosphoproteomic mass spectrometry.

Response: Thank you for your suggestion. In our revised manuscript, we updated the Introduction and References to cite the CPTAC resource for glioblastoma as follows:

“However, although several studies have conducted proteomic analysis of glioma tissue samples^{16,17} or secreted proteins in blood¹⁸, large-scale proteomic characterization in the context of GBM has not yet been reported.”

“17. The U.S. National Cancer Institute's Clinical Proteomic Tumor Analysis Consortium (CPTAC) 3 project, A Glioblastoma (GBM) discovery cohort of 99 tumor samples and 10 GTEx normal samples analyzed by global proteomic and phosphoproteomic mass spectrometry, https://pdc.cancer.gov/pdc/browse/filters/disease_type:Glioblastoma”

Reviewer #3 (Remarks to the Author):

We congratulate the authors for adequately responding to our suggestions with additional experiments and manuscript modifications.

Response: We particularly appreciate this referee's detailed and constructive suggestions in the first revision process.